# Scalable Bayesian GPFA with automatic relevance determination and discrete noise models

**Kristopher T. Jensen**[*]    **Ta-Chu Kao**[*]    **Jasmine T. Stone**    **Guillaume Hennequin**

Department of Engineering
University of Cambridge
{ktj21, tck29, jts58, gjeh2}@cam.ac.uk

## Abstract

Latent variable models are ubiquitous in the exploratory analysis of neural population recordings, where they allow researchers to summarize the activity of large populations of neurons in lower dimensional 'latent' spaces. Existing methods can generally be categorized into (i) Bayesian methods that facilitate flexible incorporation of prior knowledge and uncertainty estimation, but which typically do not scale to large datasets; and (ii) highly parameterized methods without explicit priors that scale better but often struggle in the low-data regime. Here, we bridge this gap by developing a fully Bayesian yet scalable version of Gaussian process factor analysis (bGPFA), which models neural data as arising from a set of inferred latent processes with a prior that encourages smoothness over time. Additionally, bGPFA uses automatic relevance determination to infer the dimensionality of neural activity directly from the training data during optimization. To enable the analysis of continuous recordings without trial structure, we introduce a novel variational inference strategy that scales near-linearly in time and also allows for non-Gaussian noise models appropriate for electrophysiological recordings. We apply bGPFA to continuous recordings spanning 30 minutes with over 14 million data points from primate motor and somatosensory cortices during a self-paced reaching task. We show that neural activity progresses from an initial state at target onset to a reach-specific preparatory state well before movement onset. The distance between these initial and preparatory latent states is predictive of reaction times across reaches, suggesting that such preparatory dynamics have behavioral relevance despite the lack of externally imposed delay periods. Additionally, bGPFA discovers latent processes that evolve over slow timescales on the order of several seconds and contain complementary information about reaction time. These timescales are longer than those revealed by methods which focus on individual movement epochs and may reflect fluctuations in e.g. task engagement.

## 1   Introduction

The adult human brain contains upwards of 100 billion neurons [3]. Yet many of our day-to-day behaviors such as navigation, motor control, and decision making can be described in much lower dimensional spaces. Accordingly, recent studies across a range of cognitive and motor tasks have shown that neural population activity can often be accurately summarised by the dynamics of a "latent state" evolving in a low-dimensional space [7, 8, 13, 38, 43]. Inferring and investigating these latent processes can therefore help us understand the underlying representations and computations implemented by the brain [20]. To this end, numerous latent variable models have been developed and used to analyze the activity of populations of simultaneously recorded neurons. These models range from simple linear projections such as PCA to sophisticated non-linear and temporally correlated models [10, 16, 22, 43, 50].

35th Conference on Neural Information Processing Systems (NeurIPS 2021).

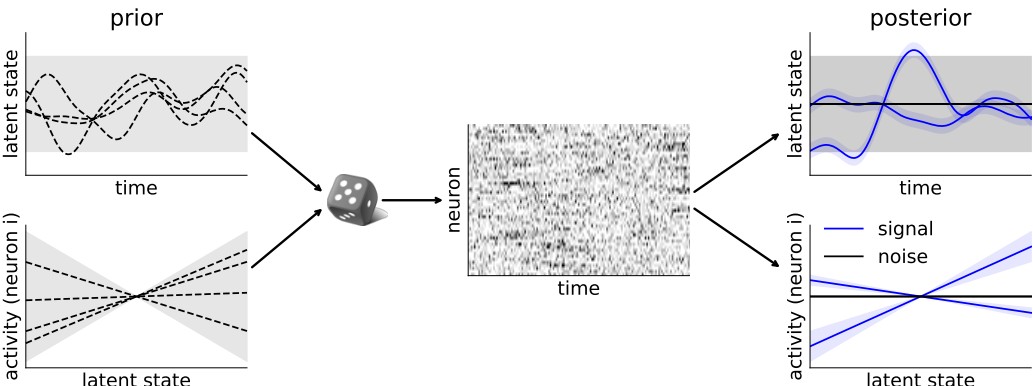

Figure 1: **Bayesian GPFA schematic.** Bayesian GPFA places a Gaussian Process prior over the latent states in each dimension as a function of time $t$ ($p(\boldsymbol{X}|\boldsymbol{t})$; top left) as well a linear prior over neural activity as a function of each latent dimension ($p(\boldsymbol{F}|\boldsymbol{X})$; bottom left). Together with a stochastic noise process $p(\boldsymbol{Y}|\boldsymbol{F})$, which can be discrete for electrophysiological recordings, this forms a generative model that gives rise to observations $\boldsymbol{Y}$ (middle). From the data and priors, bGPFA infers posterior latent states for each latent dimension ($p(\boldsymbol{X}|\boldsymbol{Y})$; top right) as well as a posterior predictive observation model for each neuron ($p(\boldsymbol{Y}_{test}|\boldsymbol{X}_{test}, \boldsymbol{Y})$; bottom right). When combined with automatic relevance determination, the model learns to automatically discard superfluous latent dimensions by maximizing the log marginal likelihood of the data (right, black vs. blue).

A popular latent variable model for neural data analysis is Gaussian process factor analysis (GPFA), which has yielded insights into neural computations ranging from time tracking to movement preparation and execution [1, 48, 49, 51]. However, fitting GPFA comes with a computational complexity of $\mathcal{O}(T^3)$ and a memory footprint $\mathcal{O}(T^2)$ for $T$ time bins. This prohibits the application of GPFA to time series longer than a few hundred time bins without artificially chunking such data into "pseudo-trials" and treating these as independent samples. Additionally, canonical GPFA assumes a Gaussian noise model while recent work has suggested that non-Gaussian models often perform better on neural data [12, 25, 60]. Here, we address these challenges by formulating a scalable and fully Bayesian version of GPFA (bGPFA; Figure 1) with a computational complexity of $\mathcal{O}(T \log T)$ and a memory cost of $\mathcal{O}(T)$. To do this, we introduce an efficiently parameterized variational inference strategy that ensures scalability to long recordings while also supporting non-Gaussian noise models. Additionally, the Bayesian formulation provides a framework for principled model selection based on approximate marginal likelihoods [53]. This allows us to perform automatic relevance determination and thus fit a single model without prior assumptions about the underlying dimensionality, which is instead inferred from the data itself [5, 41].

We validate our method on synthetic and biological data, where bGPFA exhibits superior performance to GPFA and Poisson GPFA with increased scalability and without requiring cross-validation to select the latent dimensionality. We then apply bGPFA to longitudinal, multi-area recordings from primary motor (M1) and sensory (S1) areas during a monkey self-paced reaching task spanning 30 minutes. bGPFA readily scales to such datasets, and the inferred latent trajectories improve decoding of kinematic variables compared to the raw data. This decoding improves further when taking into account the temporal offset between motor planning encoded by M1 and feedback encoded by S1. We also show that the latent trajectories for M1 converge to consistent regions of state space for a given reach direction at the onset of each individual reach. Importantly, the distance in latent space to this preparatory state from the state at target onset is predictive of reaction times across reaches, similar to previous results in a task that includes an explicit 'motor preparation epoch' where the subject is not allowed to move [1]. This illustrates the functional relevance of such preparatory activity and suggests that motor preparation takes place even when the task lacks well-defined trial structure and externally imposed delay periods, consistent with findings by Lara et al. [32] and Zimnik and Churchland [62]. Finally, we analyze the task relevance of slow latent processes identified by bGPFA which evolve on timescales of seconds; longer than the millisecond timescales that can be resolved by methods designed for trial-structured data. We find that some of these slow processes are also predictive of reaction time across reaches, and we hypothesize that they reflect task engagement which varies over the course of several reaches.

## 2 Method

In the following, we use the notation $\boldsymbol{A}$ to refer to the matrix with elements $a_{ij}$. We use $\boldsymbol{a}_k$ to refer to the k[th] row *or* column of $\boldsymbol{A}$ with an index running from 1 to $K$, represented as a column vector.

### 2.1 Generative model

Latent variable models for neural recordings typically model the neural activity $\boldsymbol{Y} \in \mathbb{R}^{N \times T}$ of $N$ neurons at times $\boldsymbol{t} \in \mathbb{R}^T$ as arising from shared fluctuations in $D$ latent variables $\boldsymbol{X} \in \mathbb{R}^{D \times T}$. Specifically, the probability of a given recording can be written as

$$p(\boldsymbol{Y}|\boldsymbol{t}) = \int p(\boldsymbol{Y}|\boldsymbol{F})\, p(\boldsymbol{F}|\boldsymbol{X})\, p(\boldsymbol{X}|\boldsymbol{t})\, d\boldsymbol{F}\, d\boldsymbol{X}, \tag{1}$$

where $\boldsymbol{F} \in \mathbb{R}^{N \times T}$ are intermediate, neuron-specific variables that can often be thought of as firing rates or a similar notion of noise-free activity. For example, GPFA [59] specifies

$$p(\boldsymbol{Y}|\boldsymbol{F}) = \prod_{n,t} \mathcal{N}(y_{nt}; f_{nt}, \sigma_n^2) \tag{2}$$

$$p(\boldsymbol{F}|\boldsymbol{X}) = \delta(\boldsymbol{F} - \boldsymbol{CX}) \tag{3}$$

$$p(\boldsymbol{X}|\boldsymbol{t}) = \prod_d \mathcal{N}(\boldsymbol{x}_d; \boldsymbol{0}, \boldsymbol{K}_d) \qquad \text{with } \boldsymbol{K}_d = k_d(\boldsymbol{t}, \boldsymbol{t})) \tag{4}$$

That is, the prior over the $d$[th] latent function $x_d(t)$ is a Gaussian process [45] with covariance function $k_d(\cdot, \cdot)$ (usually a radial basis function), and the observation model $p(\boldsymbol{Y}|\boldsymbol{X})$ is given by a parametric linear transformation with independent Gaussian noise.

In this work, we additionally introduce a prior distribution over the mixing matrix $\boldsymbol{C} \in \mathbb{R}^{N \times D}$ with hyperparameters specific to each latent dimension. This allows us to *learn* an appropriate latent dimensionality for a given dataset using automatic relevance determination (ARD) similar to previous work in Bayesian PCA (Appendix I; 5) rather than relying on cross-validation or ad-hoc thresholds of variance explained. Unlike in standard GPFA, the log marginal likelihood (Equation 1) becomes intractable with this prior. We therefore develop a novel variational inference strategy [55] which also (i) provides a scalable implementation appropriate for long continuous neural recordings, and (ii) extends the model to general non-Gaussian likelihoods better suited for discrete spike counts.

In this new framework, which we call Bayesian GPFA (bGPFA), we use a Gaussian prior over $\boldsymbol{C}$ of the form $c_{nd} \sim \mathcal{N}(0, s_d^2)$, where $s_d$ is a scale parameter associated with latent dimension $d$. Integrating $\boldsymbol{C}$ out in Equation 3 then yields the following observation model:

$$p(\boldsymbol{F}|\boldsymbol{X}) = \prod_n \mathcal{N}(\boldsymbol{f}_n; 0, \boldsymbol{X}^T \boldsymbol{S}^2 \boldsymbol{X}), \qquad \text{with } \boldsymbol{S} = \text{diag}(s_1, \ldots, s_D). \tag{5}$$

Moreover, we use a general noise model $p(\boldsymbol{Y}|\boldsymbol{F}) = \prod_{n,t} p(y_{nt}|f_{nt})$ where $p(y_{nt}|f_{nt})$ is any distribution for which we can evaluate its density.

### 2.2 Variational inference and learning

To train the model and infer both $\boldsymbol{X}$ and $\boldsymbol{F}$ from the data $\boldsymbol{Y}$, we use a nested variational approach. It is intractable to compute $\log p(\boldsymbol{Y}|\boldsymbol{t})$ (Equation 1) analytically for bGPFA, and we therefore introduce a lower bound on $\log p(\boldsymbol{Y}|\boldsymbol{t})$ at the outer level and another one on $\log p(\boldsymbol{Y}|\boldsymbol{X})$ at the inner level. These lower bounds are constructed from approximations to the posterior distributions over latents ($\boldsymbol{X}$) and noise-free activity ($\boldsymbol{F}$) respectively.

**Distribution over latents**  At the outer level, we introduce a variational distribution $q(\boldsymbol{X})$ over latents and construct an evidence lower bound (ELBO; 55) on the log marginal likelihood of Equation 1:

$$\log p(\boldsymbol{Y}|\boldsymbol{t}) \geq \mathcal{L} := \mathbb{E}_{q(\boldsymbol{X})}\left[\log p(\boldsymbol{Y}|\boldsymbol{X})\right] - \text{KL}\left[q(\boldsymbol{X})||p(\boldsymbol{X}|\boldsymbol{t})\right]. \tag{6}$$

Conveniently, maximizing this lower bound is equivalent to minimizing $\text{KL}\left[q(\boldsymbol{X})||p(\boldsymbol{X}|\boldsymbol{Y})\right]$ and thus also yields an approximation to the posterior over latents in the form of $q(\boldsymbol{X})$. We estimate the first term of the ELBO using Monte Carlo samples from $q(\boldsymbol{X})$ and compute the KL term analytically.

Here, we use a so-called whitened parameterization of $q(\boldsymbol{X})$ [19] that is both expressive and scalable to large datasets:

$$q(\boldsymbol{X}) = \prod_{d=1}^{D} \mathcal{N}(\boldsymbol{x}_d; \boldsymbol{\mu}_d, \boldsymbol{\Sigma}_d) \quad \text{with} \quad \boldsymbol{\mu}_d = \boldsymbol{K}_d^{\frac{1}{2}} \boldsymbol{\nu}_d \quad \text{and} \quad \boldsymbol{\Sigma}_d = \boldsymbol{K}_d^{\frac{1}{2}} \boldsymbol{\Lambda}_d \boldsymbol{\Lambda}_d^T \boldsymbol{K}_d^{\frac{1}{2}^T}, \quad (7)$$

where $\boldsymbol{K}_d^{\frac{1}{2}}$ is any square root of the prior covariance matrix $\boldsymbol{K}_d$, and $\boldsymbol{\nu}_d \in \mathbb{R}^T$ is a vector of variational parameters to be optimized. $\boldsymbol{\Lambda}_d \in \mathbb{R}^{T \times T}$ is a positive semi-definite variational matrix whose structure is chosen carefully so that its squared Frobenius norm, log determinant, and matrix-vector products can all be computed efficiently which facilitates the evaluation of Equations 8 and 9. This whitened parameterization has several advantages. First, it does not place probability mass where the prior itself does not. In addition to stabilizing learning [39], this also guarantees that the posterior is temporally smooth for a smooth prior. Second, the KL term in Equation 6 simplifies to

$$\text{KL}[q(\boldsymbol{X})||p(\boldsymbol{X}|\boldsymbol{t})] = \frac{1}{2} \sum_d \left( ||\boldsymbol{\Lambda}_d||_\text{F}^2 - 2 \log |\boldsymbol{\Lambda}_d| + ||\boldsymbol{\nu}_d||^2 - T \right). \quad (8)$$

Third, $q(\boldsymbol{X})$ can be sampled efficiently via a differentiable transform (i.e. the reparameterization trick) provided that fast differentiable $\boldsymbol{K}_d^{\frac{1}{2}} \boldsymbol{v}$ and $\boldsymbol{\Lambda}_d \boldsymbol{v}$ products are available for any vector $\boldsymbol{v}$:

$$\boldsymbol{x}_d^{(m)} = \boldsymbol{K}_d^{\frac{1}{2}} (\boldsymbol{\nu}_d + \boldsymbol{\Lambda}_d \boldsymbol{\eta}_d) \quad \text{with} \quad \boldsymbol{\eta}_d \sim \mathcal{N}(\boldsymbol{0}, \boldsymbol{I}), \quad (9)$$

where $\boldsymbol{x}_d^{(m)} \sim q(\boldsymbol{x}_d)$. This is important to form a Monte Carlo estimate of $\mathbb{E}_{q(\boldsymbol{X})}[\log p(\boldsymbol{Y}|\boldsymbol{X})]$.

To avoid the challenging computation of $\boldsymbol{K}_d^{\frac{1}{2}} \boldsymbol{v}$ for general $\boldsymbol{K}_d$ [2], we directly parameterize $\boldsymbol{K}_d^{\frac{1}{2}}$, the positive definite square root of $\boldsymbol{K}$, which implicitly defines the prior covariance function $k_d(\cdot, \cdot)$. In this work we use an RBF kernel for $\boldsymbol{K}_d$ and give the expression for $\boldsymbol{K}_d^{\frac{1}{2}}$ in Appendix F. Additionally, we use Toeplitz acceleration methods to compute $\boldsymbol{K}_d^{\frac{1}{2}} \boldsymbol{v}$ products in $\mathcal{O}(T \log T)$ time and with $\mathcal{O}(T)$ memory cost [48, 57].

We implement and compare different choices of $\boldsymbol{\Lambda}_d$ in Appendix F. For the experiments in this work, we use the parameterization $\boldsymbol{\Lambda}_d = \boldsymbol{\Psi}_d \boldsymbol{C}_d$, where $\boldsymbol{\Psi}_d$ is diagonal with positive entries and $\boldsymbol{C}_d$ is circulant, symmetric, and positive definite. This parameterization enables cheap computation of KL divergences and matrix-vector products while maintaining sufficient expressiveness (Appendix F). All results are qualitatively similar when instead using a simple diagonal parameterization $\boldsymbol{\Lambda}_d = \boldsymbol{\Psi}_d$.

**Distribution over neural activity**  Evaluating $\log p(\boldsymbol{Y}|\boldsymbol{X}) = \sum_n \log p(\boldsymbol{y}_n|\boldsymbol{X})$ for each sample drawn from $q(\boldsymbol{X})$ is intractable for general noise models. Thus, we further lower-bound the ELBO of Equation 6 by introducing an approximation $q(\boldsymbol{f}_n|\boldsymbol{X})$ to the posterior $p(\boldsymbol{f}_n|\boldsymbol{y}_n, \boldsymbol{X})$:

$$\log p(\boldsymbol{y}_n|\boldsymbol{X}) \geq \mathbb{E}_{q(\boldsymbol{f}_n|\boldsymbol{X})}[\log p(\boldsymbol{y}_n|\boldsymbol{f}_n)] - \text{KL}\left[q(\boldsymbol{f}_n|\boldsymbol{X})||p(\boldsymbol{f}_n|\boldsymbol{X})\right]. \quad (10)$$

We repeat the whitened variational strategy described at the outer level by writing

$$q(\boldsymbol{f}_n|\boldsymbol{X}) = \mathcal{N}(\boldsymbol{f}_n; \hat{\boldsymbol{\mu}}_n, \hat{\boldsymbol{\Sigma}}_n) \quad \text{with} \quad \hat{\boldsymbol{\mu}}_n = \hat{\boldsymbol{K}}^{\frac{1}{2}} \hat{\boldsymbol{\nu}}_n \quad \text{and} \quad \hat{\boldsymbol{\Sigma}}_n = \hat{\boldsymbol{K}}^{\frac{1}{2}} \boldsymbol{L}_n \boldsymbol{L}_n^T (\hat{\boldsymbol{K}}^{\frac{1}{2}})^T, \quad (11)$$

where $\hat{\boldsymbol{\nu}}_n \in \mathbb{R}^D$ is a neuron-specific vector of variational parameters to be optimized along with a lower-triangular matrix $\boldsymbol{L}_n \in \mathbb{R}^{D \times D}$; and $\hat{\boldsymbol{K}}$ denotes the covariance matrix of $p(\boldsymbol{f}|\boldsymbol{X})$, whose square root $\hat{\boldsymbol{K}}^{\frac{1}{2}} = \boldsymbol{X}^T \boldsymbol{S}$ follows from Equation 5. The low-rank structure of $\hat{\boldsymbol{K}}$ enables cheap matrix-vector products and KL divergences:

$$\text{KL}[q(\boldsymbol{f}_n|\boldsymbol{X})||p(\boldsymbol{f}_n|\boldsymbol{X})] = \frac{1}{2} \left( ||\boldsymbol{L}_n||_\text{F}^2 - 2 \log |\boldsymbol{L}_n| + ||\hat{\boldsymbol{\nu}}_d||^2 - D \right). \quad (12)$$

Note that the KL divergence does not depend on $\boldsymbol{X}$ in this whitened parameterization (Appendix H). Moreover, $q(\boldsymbol{f}_n|\boldsymbol{X})$ in Equation 11 has the form of the exact posterior when the noise model is Gaussian (Appendix G), and it is equivalent to a stochastic variational inducing point approximation [18] for general noise models (Appendix H).

Finally, we need to compute the first term in Equation 10:

$$\mathbb{E}_{q(\boldsymbol{f}_n|\boldsymbol{X})}[\log p(\boldsymbol{y}_n|\boldsymbol{f}_n)] = \sum_t \mathbb{E}_{q(f_{nt}|\boldsymbol{X})}[\log p(y_{nt}|f_{nt})]. \quad (13)$$

Each term in this sum is simply a 1-dimensional Gaussian expectation which can be computed analytically in the case of Gaussian or Poisson noise (with an exponential link function), and otherwise approximated efficiently using Gauss-Hermite quadrature (Appendix K; [18]).

## 2.3 Summary of the algorithm

Putting Section 2.1 and Section 2.2 together, optimization proceeds at each iteration by drawing $M$ Monte Carlo samples $\{\boldsymbol{X}_m\}_1^M$ from $q(\boldsymbol{X})$ and estimating the overall ELBO as:

$$\mathcal{L} = \frac{1}{M} \sum_{\boldsymbol{X}_m \sim q(\boldsymbol{X})} \left[ \sum_{n,t} \mathbb{E}_{q(f_{nt}|\boldsymbol{X}_m)} \left[ \log p(y_{nt}|f_{nt}) \right] \right]$$
$$- \sum_n \mathrm{KL}\left[ q(\boldsymbol{f}_n) || p(\boldsymbol{f}_n) \right] - \sum_d \mathrm{KL}\left[ q(\boldsymbol{x}_d) || p(\boldsymbol{x}_d) \right], \quad (14)$$

where the expectation over $q(f_{nt}|\boldsymbol{X})$ is evaluated analytically or using Gauss-Hermite quadrature depending on the noise model (Appendix K). We maximize $\mathcal{L}$ using stochastic gradient ascent with Adam [29]. This has a total computational time complexity of $\mathcal{O}(MNTD^2 + MDT \log T)$ and memory complexity of $\mathcal{O}(MNTD^2)$ where $N$ is the number of neurons, $T$ the number of time points, and $D$ the latent dimensionality. For large datasets such as the monkey reaching data in Section 3.2, we compute gradients using mini-batches across time to mitigate the memory cost. That is, gradients for the sum over $t$ in Equation 14 are computed in multiple passes. The algorithm is described in pseudocode with further implementation and computational details in Appendix L. The model learned by bGPFA can subsequently be used for predictions on held-out data by conditioning on partial observations as used for cross-validation in Section 3.1 and discussed in Appendix M. Latent dimensions that have been 'discarded' by automatic relevance determination will automatically have negligible contributions to the resulting posterior predictive distribution since the prior scale parameters $s_d$ are approximately zero for these dimensions (see Appendix I for details).

# 3 Experiments and results

## 3.1 Synthetic data

We first generated an example dataset from the GPFA generative model (Equations 2-4) with a true latent dimensionality of 3. We proceeded to fit both factor analysis (FA), GPFA, and bGPFA with different latent dimensionalities $D \in [1, 10]$. Here, we fitted bGPFA without automatic relevance determination such that $s_d = s \, \forall d$. As expected, the marginal likelihoods increased monotonically with $D$ for both FA and GPFA (Figure 2a; Appendix I). In contrast, the bGPFA ELBO reached its optimum value at the true latent dimensionality $D^\star = 3$. This is a manifestation of "Occam's razor", whereby fully Bayesian approaches favor the simplest model that adequately explains the data $\boldsymbol{Y}$ [36]. When instead considering the cross-validated predictive performance of each method, performance deteriorated rapidly for $D > 3$ for FA and GPFA, while Bayesian GPFA was more robust to overfitting (Figure 2b). Notably, the introduction of ARD parameters $\{s_d\}$ in bGPFA allowed us to fit a single model with large $D = 10$. This recovered the maximum ELBO of bGPFA without ARD and the minimum test error across GPFA and bGPFA without ARD (Figure 2a and b, black) without *a priori* assumptions about the latent dimensionality or the need to perform extensive cross-validation. Consistent with the ground truth generative process, only 3 of the scale parameters $s_d$ remained well above zero after training (Figure 2b, inset). Similar to this illustrative example with Gaussian data, bGPFA with ARD and Poisson noise also exceeded the optimal performance of Poisson GPFA when applied to both synthetic and experimental spike count data (Appendix E).

We then proceeded to apply bGPFA ($D = 10$) to an example dataset drawn using Equations 4 and 5 with a ground truth dimensionality $D^\star = 2$, and either Gaussian, Poisson, or negative binomial noise. For all three datasets, the learned parameters clustered into a group of two latent dimensions with high information content (Appendix J) and a group of eight uninformative dimensions, consistent with the generative process (Figure 2c). In each case, we extracted the inferred latent trajectories corresponding to the informative dimensions and found that they recapitulated the ground truth up to a linear transformation (Figure 2d). Fitting flexible noise models such as the negative binomial model is important because neural firing patterns are known to be overdispersed in many contexts [4, 15, 54]. However, it is often unclear how much of that overdispersion should be attributed to common fluctuations in hidden latent variables ($\boldsymbol{X}$ in our model) compared to private noise processes in single neurons [35]. In our synthetic data with negative binomial noise, we could accurately recover the single-neuron overdispersion parameters (Figure 2e; Appendix K), suggesting that such unsupervised models have the capacity to resolve overdispersion due to private and shared processes.

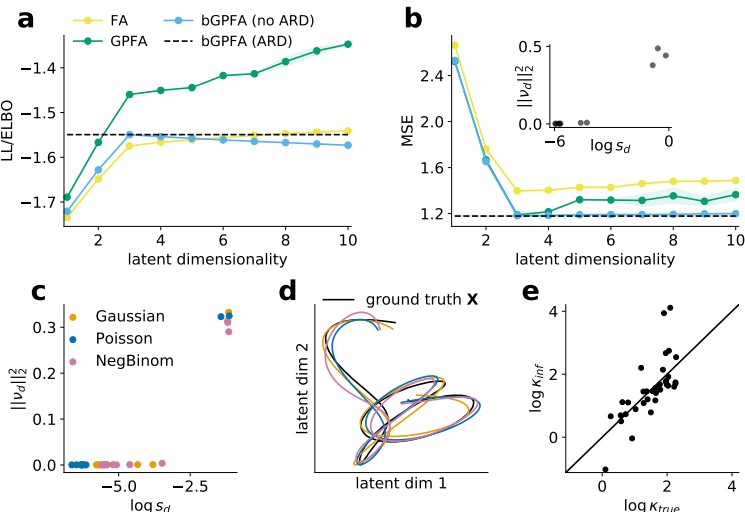

Figure 2: **Bayesian GPFA applied to synthetic data.** **(a)** Log likelihoods of factor analysis (yellow) & GPFA (green) and ELBO of Bayesian GPFA without ARD (blue) fitted to synthetic data with a ground truth dimensionality of three for different model dimensionalities. bGPFA with ARD recovered a three-dimensional latent space as well as the optimum ELBO of bGPFA without ARD (black dashed line). **(b)** Cross-validated prediction errors for the models in (a) (Appendix M). bGPFA with ARD recovered the performance of the optimal GPFA and bGPFA models without requiring a search over latent dimensionalities. Inspection of the learned prior scales $\{s_d\}$ and posterior mean parameters $||\nu_d||_2^2$ (inset) indicates that ARD retained only $D^\star = 3$ informative dimensions (top right) and discarded the other 7 dimensions (bottom left). Shadings in (a) and (b) indicate $\pm 2$ stdev. across 10 model fits. **(c)** Learned parameters of bGPFA with ARD and either Gaussian, Poisson or negative binomial noise models fitted to two-dimensional synthetic datasets with observations drawn from the corresponding noise models (Appendix K). The parameters clustered into two groups of informative (top right) and non-informative (bottom left) dimensions (Appendix J). **(d)** Latent trajectory in the space of the two most informative dimensions (c.f. (c)) for each model with the ground truth shown in black. **(e)** The overdispersion parameter $\kappa_n$ for each neuron learned in the negative binomial model, plotted against the ground truth (Appendix K). Solid line indicates $y = x$; note that $\kappa_n \to \infty$ corresponds to a Poisson noise model.

In summary, bGPFA provides a flexible method for inferring both latent dimensionalities, latent trajectories, and heterogeneous single-neuron parameters in an unsupervised manner. In the next section, we show that the scalability of the model and its interpretable parameters also facilitate the analysis of large neural population recordings.

## 3.2 Primate recordings

In this section, we apply bGPFA to biological data recorded from a rhesus macaque during a self-paced reaching task with continuous recordings spanning 30 minutes (37, 42; Figure 3a). The continuous nature of these recordings as one long trial makes it a challenging dataset for existing analysis methods that explicitly require the availability of many trials per experimental condition [43], and poses computational challenges to Gaussian process-based methods that cannot handle long time series [59]. While the ad-hoc division of continuous recordings into surrogate trials can still enable the use of these methods [28], here we show that our formulation of bGPFA readily applies to long continuous recordings. We fitted bGPFA with a negative binomial noise model to recordings from both primary motor cortex (M1) and primary somatosensory cortex (S1). For all analyses, we used a single recording session (`indy_20160426`, as in 28), excluded neurons with overall firing rates below 2 Hz, and binned data at 25 ms resolution. This resulted in a data array $Y \in \mathbb{R}^{200 \times 70482}$ (130 M1 neurons and 70 S1 neurons).

We first fitted bGPFA independently to the M1 and S1 sub-populations with $D = 25$ latent dimensions. In this case, ARD retained 16 (M1) and 12 (S1) dimensions (Figure 3b). We then proceeded to train a linear decoder to predict hand kinematics in the form of $x$ and $y$ hand velocities from either

the inferred firing rates or the raw data convolved with a 50 ms Gaussian kernel ([28]; Appendix M). We found that the model learned by bGPFA predicted kinematics better than the convolved spike trains, suggesting that (i) the latent space accurately captures kinematic representations, and (ii) the denoising and data-sharing across time in bGPFA aids decodability beyond simple smoothing of neural activity. Interestingly, by repeating this decoding analysis with an artificially imposed delay between neural activity and decoded behavior, we found that neurons in S1 predominantly encoded current behavior while neurons in M1 encoded a motor plan that predicted kinematics 100-150 ms into the future (Figure 3b). This is consistent with the motor neuroscience literature suggesting that M1 functions as a dynamical system driving behavior via downstream effectors [8].

We then fitted bGPFA to the entire dataset including both M1 and S1 neurons. In this case, bGPFA retained 19 dimensions (Appendix D), and kinematic predictions improved over individual M1- and S1-based predictions (Figure 3b). In this analysis, the decoding performance as a function of delay between neural activity and behavior exhibited a broader peak than for the single-region decoding. We hypothesized that this broad peak reflects the fact that these neural populations encode both *current* behavior in S1 as well as *future* behavior in M1 (Figure 3c). Indeed, when we took this offset into account by shifting all M1 spike times by +100 ms and retraining the model, decoding performance increased from $68.56\% \pm 0.09$ to $69.81\% \pm 0.06$ (mean $\pm$ sem variance explained across ten model fits; Appendix M). Additionally, the shifted data exhibited a narrower decoding peak attained for near-zero delay between kinematics and latent trajectories (Figure 3d). Consistent with the improved kinematic decoding, we also found that shifting the M1 spikes by 100 ms increased the ELBO per neuron ($-34,637.0 \pm 0.7$ to $-34,631.1 \pm 0.6$) and decreased the dimensionality of the data (Appendix D; [46]). These results suggest that M1 and S1 contain both overlapping but also non-redundant information, and that the most parsimonious description of the neural data is recovered by taking into account the different biological properties of M1 and S1.

We next wondered if bGPFA could be used to reveal putative motor preparation processes, which is non-trivial due to the lack of trial structure and well-defined preparatory epochs. We partitioned the data post-hoc into individual 'reaches', each consisting of a period of time where the target location remained constant. For these analyses, we only considered 'successful' reaches, where the monkey eventually moved to the target location, and we defined movement onset as the first time during a reach where the cursor speed exceeded a low threshold (Appendix A). We began by visualizing the latent processes inferred by bGPFA as they unfolded prior to movement onset in each reach epoch. For visualization purposes, we ranked the latent dimensions based on their learned prior scales (a measure of variance explained; Appendix J) and selected the first two. Prior to movement onset, the latent trajectories tended to progress from their initial location at target onset towards reach-specific regions of state space (see example trials in Figure 3e for leftward and rightward reaches). To quantify this phenomenon, we computed pairwise similarities between latent states across all 762 reaches, during (i) stimulus onset and (ii) 75 ms before movement onset (chosen such that it is well before any detectable movement; Appendix A). We defined similarity as the negative Euclidean distance between latent states and restricted the analysis to 'fast' latent dimensions with timescales smaller than 200 ms to study this putatively fast process. When plotted as a function of reach direction, the latent similarities at target onset showed little discernable structure (Figure 3f, left). In contrast, the pairwise similarities became strongly structured 75 ms before movement onset where neighboring reach directions were associated with similar preparatory latent states (Figure 3f, right). Similar albeit noisier results were found when using factor analysis or GPFA instead of bGPFA (Appendix A). These findings are consistent with previous reports of monkey M1 partitioning preparatory and movement-related activity into distinct subspaces [14, 32], as well as with the analogous finding that a 'relative target' subspace is active before a 'movement subspace' in previous analyses of this particular dataset [28].

Previous work on delayed reaches has shown that monkeys start reaching earlier when the neural state attained at the time of the go cue – which marks the end of a delay period with a known reach direction – is close to an "optimal subspace" [1, 23]. We wondered if a similar effect takes place during continuous, self-initiated reaching in the absence of explicit delay periods. Based on Figure 3e, we hypothesized that the monkey should start moving earlier if, at the time the next target is presented, its latent state is already close to the mean preparatory state for the required next movement direction. To test this, we extracted the mean preparatory state 75 ms prior to movement onset (as above) for each reach direction in the dataset. We found that the distance between the latent state at target onset and the corresponding mean preparatory state was strongly predictive of reaction time (Figure 3g,

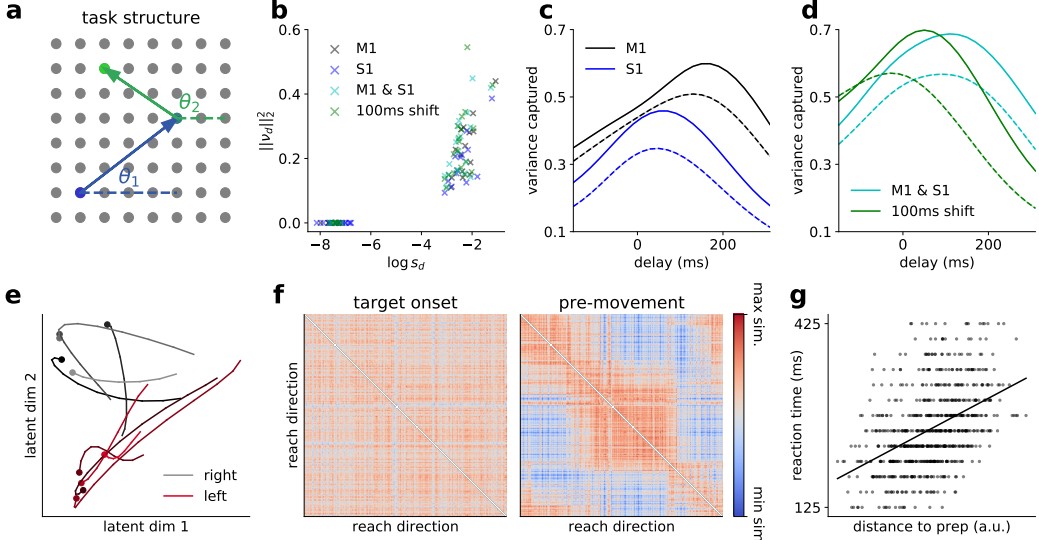

Figure 3: **Bayesian GPFA applied to primate data. (a)** Schematic illustration of the self-paced reaching task. When a target on a 17x8 grid is reached (arrows; 8x8 shown for clarity), a new target lights up on the screen (colours), selected at random from the remaining targets. In several analyses, we classify movements according to reach angle measured relative to horizontal ($\theta_1$, $\theta_2$). **(b)** Learned mean and scale parameters for the bGPFA models. Small prior scales $s_d$ and posterior mean parameters ($||\nu_d||_2^2$) indicate uninformative dimensions (Appendix J). **(c)** We applied bGPFA to monkey M1 and S1 data during the task and trained a linear model to decode kinematics from firing rates predicted from the inferred latent trajectories with different delays between latent states and kinematics. Neural activity was most predictive of future behavior in M1 (black) and current behavior in S1 (blue). Dashed lines indicate decoding from the raw data convolved with a Gaussian filter. **(d)** Decoding from bGPFA applied to the combined M1 and S1 data (cyan). Performance improved further when decoding from latent trajectories inferred from data where M1 activity was shifted by 100 ms relative to S1 activity (green). **(e)** Example trajectories in the two most informative latent dimensions for five rightward reaches (grey) and five leftward reaches (red). Trajectories are plotted from the appearance of the stimulus until movement onset (circles). During 'movement preparation', the latent trajectories move towards a consistent region of latent state space for each reach direction. **(f)** Similarity matrix of the latent state at stimulus onset showing no obvious structure (left) and 75 ms prior to movement onset showing modulation by reach direction (right). **(g)** Reaction time plotted against Euclidean distance between the latent state at target onset and the mean preparatory state for the corresponding reach direction ($\rho = 0.45$).

Pearson $\rho = 0.45$, $p = 4 \times 10^{-36}$). Such a correlation was also weakly present with factor analysis ($\rho = 0.21$, $p = 1.1 \times 10^{-8}$) but not detectable in the raw data ($\rho = 0.002$, $p = 0.95$). We also verified that the strong correlation found with bGPFA was not an artifact of the temporal correlations introduced by the prior (Appendix C). Taken together, our results suggest that motor preparation is an important part of reaching movements even in an unconstrained self-paced task. Additionally, we showed that bGPFA captures such behaviorally relevant latent dynamics better than simpler alternatives, and our scalable implementation enables its use on the large continuous reaching dataset analysed here.

### 3.3 Long-timescale latent processes

Some latent dimensions inferred by bGPFA also had long timescales on the order of 1.5 seconds, which is similar to the timescale of individual reaches (1-2 seconds; Appendix C). We hypothesized that these slow dynamics might reflect motivation or task engagement. Consistent with this hypothesis, we found that one of the slow latent processes ($\tau = 1.4$ s) was strongly correlated with reaction time during successful reaches (Pearson $\rho = 0.40$, $p = 3.4 \times 10^{-28}$). Interestingly, the information contained about reaction time in this long timescale latent dimension was largely complementary to that encoded by the distance to preparatory states in the 'fast' dimensions (Appendix C), suggesting

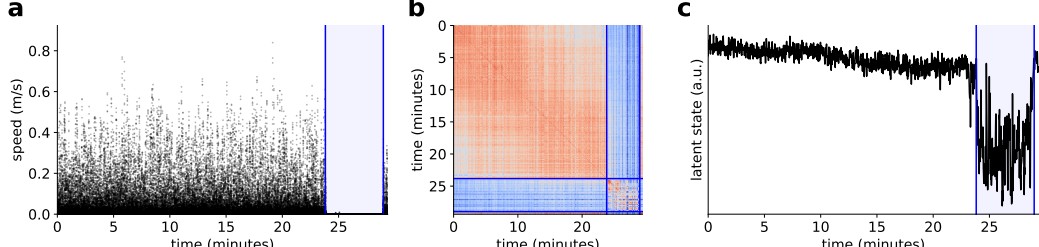

Figure 4: **Analysis of a period without task participation.** **(a)** Cursor speed over the course of the recording session. Blue horizontal lines indicate the last successful trial before and first successful trial after a period with no active task participation (blue shading). **(b)** Latent similarity matrix as a function of time during the task. The latent dynamics during task participation occur in a largely orthogonal subspace to the dynamics during the period with no active task participation. **(c)** Plot of latent state over time for a long-timescale latent dimension strongly correlated with reaction time ($\tau = 1.4$ s).

that motor preparation and task engagement are orthogonal processes both contributing to task performance.

The experimental recordings were also characterized by a period of approximately five minutes towards the end of the recording session during which the monkey did not participate actively in the task and the cursor velocity was near-constant at zero (Figure 4a). When analysing neural activity across the periods with and without task participation, we found that neural dynamics moved to a different subspace as the monkey stopped engaging with the task (Figure 4b). Importantly, we were able to simultaneously capture these context-dependent changes as well as movement-specific and preparatory dynamics (Section 3.2) by fitting a single model to the full 30 minute dataset. This suggests that bGPFA can capture behaviorally relevant dynamics within individual contexts even when trained on richer datasets with changing contexts.

Finally, we wondered how the neural activity patterns during periods with and without task participation were related to the long-timescale latent dimensions predictive of task engagement. Here we found that the slow latent process considered above also exhibited a prominent change to a different state as the monkey stopped participating in the task (Figure 4). This is consistent with our hypothesis that this latent process captures a feature related to task engagement which slowly deteriorated during the first 24 minutes of the task followed by a discrete switch to a state with no engagement in the task. During the period of active task participation, this latent dimension was also correlated with time within the session. Indeed, reach number and latent state were both predictive of reaction time, but with the latent trajectory exhibiting a slightly stronger correlation (Pearson $\rho = 0.40$ vs. $\rho = 0.37$). It is not surprising that task engagement decreases with time, and it is in this case difficult to tease apart how motivation and time are differentially represented in such latent processes. However, based on the strong and abrupt modulation by task participation, this latent dimension appears to represent an aspect of engagement with the task beyond the passing of time.

Taken together, we thus find that bGPFA is capable of capturing not only single-reach dynamics and preparatory activity but also complementary processes evolving over longer timescales, which would be difficult to identify with methods designed for the analysis of many shorter trials.

## 4 Discussion

**Related work** The generative model of bGPFA can be considered an extension of the canonical GPFA model proposed by Yu et al. [59] to include a Gaussian prior over the loading matrix $C$ (Section 2.1). In this view, bGPFA is to GPFA what Bayesian PCA is to PCA [5]; in particular, it facilitates automatic relevance determination to infer the dimensionality of the latent space from data [5, 41, 53]. Similar to previous work in the field, we also use variational inference to facilitate arbitrary observation noise models, including non-Gaussian models more appropriate for electrophysiological recordings [12, 25, 34, 50, 60, 61]. While variational inference has proven a useful framework for such non-conjugate likelihood models, alternative approaches exist including the use of polynomial approximations to the non-linear terms in the likelihood [24]. Another major challenge in the

development of GP-based latent variable models such as bGPFA is to ensure scalability for longer time series. In this work, we utilize advances in variational inference [30, 47] to facilitate scalability to the large datasets recorded in modern neuroscience. In particular, we contribute a new circulant variational GP posterior expressed partly in the Fourier domain that is both accurate and scalable. This is similar to Keeley et al. [25], who address the problem of scalability by assuming independence across Fourier features and formulating variational inference in the Fourier domain. However, we instead perform inference in the time domain and include additional factors in our variational posterior that ensure smoothness over time and allow for non-stationary posterior covariances. In contrast to these approaches, Zhao and Park [60] rely on a low rank approximation to the prior covariance for inference and temporal subsamples for hyperparameter optimization to overcome the computational cost of model training. A conceptually similar approach employed by Duncker and Sahani [12] is the use of inducing points which has been studied extensively in the Gaussian process literature [17, 18, 52]. However, such low rank approximations can perform poorly on long time series where the number of inducing points needed is proportional to the recording duration [6].

bGPFA is also closely related to Gaussian process latent variable models (GPLVMs) [33, 53] which have recently found use in the neuroscience literature as a way of modelling flexible, nonlinear tuning curves [22, 34, 58]. This is because integrating out the loading matrix $C$ in $p(Y|X)$ with a Gaussian prior gives rise to a Gaussian process with a linear kernel. The low-rank structure of this linear kernel yields computationally cheap likelihoods, and our variational approach to estimating $\log p(Y|X)$ is in fact equivalent to the sparse inducing point approximation used in the stochastic variational GP (SVGP) framework [17, 18]. In particular, our variational posterior is the same as that which would arise in SVGP with at least $D$ inducing points irrespective of where those inducing points are placed (Appendix H). We also note that for a Gaussian noise model, the resulting low-rank Gaussian posterior is the form of the exact posterior distribution (Appendix G). Additionally, since the bGPFA observation model and prior over latents are both GPs, bGPFA is an example of a deep GP [11] with two layers – the first with an RBF kernel and the second with a linear kernel. Finally, our parameterizations of the posteriors $q(x_d)$ and $q(f_n)$ can be viewed as variants of the 'whitening' approach introduced by Hensman et al. [19] which both facilitates efficient computation of the KL terms in the ELBOs and also stabilizes training (Section 2.2).

**Conclusion & impact**   In summary, bGPFA is an extension of the popular GPFA model in neuroscience that allows for regularized, scalable inference and automatic determination of the latent dimensionality as well as the use of non-Gaussian noise models appropriate for neural recordings. Importantly, the hyperparameters of bGPFA are efficiently optimized based on the ELBO on training data, which alleviates the need for cross-validation or complicated algorithms otherwise used for hyperparameter optimization in overparameterized models [16, 22, 27, 28, 58, 59]. Our approach can also be extended in several ways to make it more useful to the neuroscience community. For example, replacing the spike count-based noise models with a point process model would provide higher temporal resolution [12], and facilitate inference of optimal temporal delays across neural populations [31]. This will likely be useful as multi-region recordings become more prevalent in neuroscience [26]. Additionally, by substituting the linear kernel in $p(Y|X)$ for an RBF kernel in Euclidean space [58] or on a non-Euclidean manifold [22], we can recover scalable versions of recent GPLVM-based tools for neural data analyses with automatic relevance determination.

While such latent variable models have already played a major role in systems neuroscience, progress in the field is still hampered by the relative inaccessability of these methods to the general neuroscience community. Here, we provide a ready-to-use Python package with GPU implementations of not only bGPFA with ARD, but also standard GPFA and Factor Analysis with Gaussian and non-Gaussian noise models. We hope that this implementation will help bridge the gap between methods development and the practical use of these methods by the community, similar to other recent efforts to democratize and compare latent variable models for neuroscience [44]. In future work, it will also be interesting to extend bGPFA to non-linear observation models $p(Y|X)$ and provide a general-purpose package for latent variable modelling in neuroscience. As progress in neuroscience continues to accelerate, these methods are likely to find extensive use in medical and industrial settings with applications e.g. to brain-computer interfaces in human subjects [21, 40, 56]. With the rise of such technology transfer, the research community should strive to maintain research transparency by providing open source code for methods development, and to remain aware of ethical issues associated with the increasingly narrow gap between research and applications in humans and other animals [9].

## Acknowledgements

We are grateful to O'Doherty et al. [42] for making their data publicly available and to Marine Schimel and David Liu for insightful discussions. We thank Marine Schimel, Yashar Ahmadian, Peter Stone, and Jonathan So for helpful comments on the manuscript. We thank David Liu for contributions to the codebase used for our analyses. K.T.J. was funded by a Gates Cambridge scholarship and J.T.S. by a Churchill scholarship.

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
