# Appendix – Scalable Bayesian GPFA with automatic relevance determination and discrete noise models

## A    Further analyses of preparatory dynamics in the primate reaching task

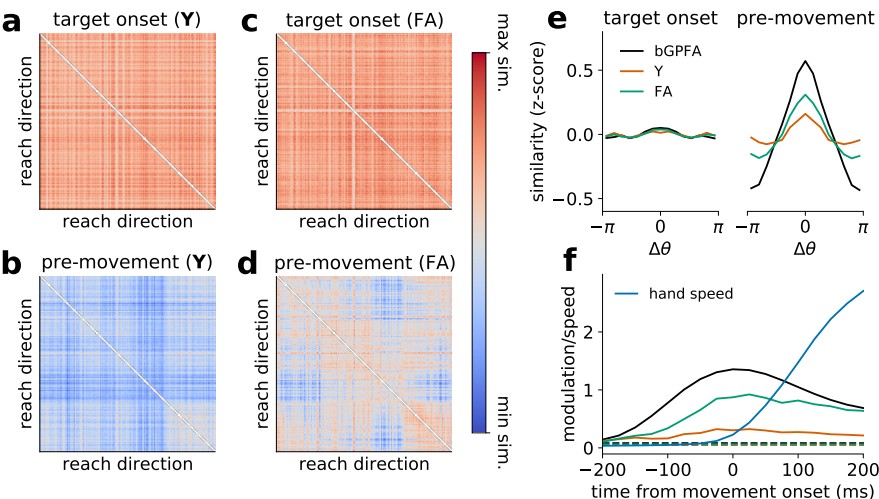

Figure 5: **Further analyses of M1 preparatory dynamics. (a-d)** Similarity matrix of raw neural activity $Y$ (a & b) and latent states found by FA (c & d) at target onset (a & c) and 75 ms prior to movement onset (b & d), with analyses performed as in Figure 3f. **(e)** z-scored similarity as a function of difference in reach direction; here, the mean similarity across pairs of reaches is shown at target onset (left) and 75 ms prior to movement onset (right). The bGPFA latent states show much stronger modulation than either raw neural activity ($Y$) or latent states from FA. **(f)** Modulation of similarity by reach direction as a function of time from movement onset. Modulation was defined as the difference between maximum and minimum z-scored similarity as a function of difference in reach direction (peak-to-trough in panel e). Blue solid line indicates the z-scored hand speed, confirming the absence of premature movement relative to our definition of movement onset. bGPFA latent similarity increases well before hand speed and starts decreasing substantially before the hand speed peaks. Dashed lines indicate modulation at target onset for each method.

We performed analyses as in Figure 3f using the raw data ($Y$) and using factor analysis (FA) with 20 latent dimensions instead of using the bGPFA latent states. The raw data $Y$ showed a high degree of similarity at target onset compared to movement onset, but little discernable structure as a function of reach direction at either point in time (Figure 5a-b).

While the FA latent distances exhibited no modulation by reach direction at target onset, FA did discover weak modulation at movement onset (Figure 5a-b). This is qualitatively consistent with our results using bGPFA but with a lower signal to noise ratio. Here and in Section 3.2, we defined movement onset as the first time during a reach where the cursor velocity exceeded $0.025\,\mathrm{m\,s^{-1}}$, and we observed little to no quantifiable movement before this point (Figure 5f). We also discarded 'trials' with premature movement for all analyses here and in Section 3.2, which we defined as reaches with a reaction time of 75 ms or less.

To quantify and compare how neural activity was modulated by the similarity of reach directions for different analysis methods, we first computed z-scores of the similarity matrices for both the bGPFA latent states, raw activity, and the latent states from FA. z-scores were calculated as $z = (S - mean(S))/std(S)$ for each similarity matrix $S$, and the diagonal elements were excluded for this analysis. We then computed the mean of the z-scored pairwise similarities as a function of difference in reach direction across all pairs of 762 reaches. We found that none of the datasets

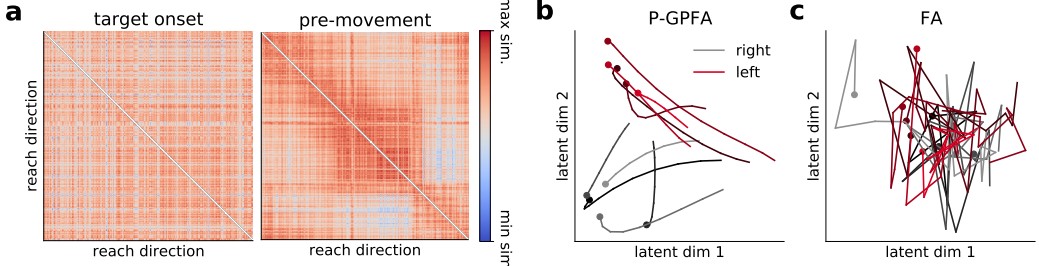

Figure 6: **Analyses of M1 dynamics with GPFA and FA. (a)** P-GPFA was fitted to data recorded from M1 during the self-paced reaching task. We computed the similarity matrix of the latent state at stimulus onset showing no obvious structure (left) and 75 ms prior to movement onset showing modulation by reach direction (right). Reaches are sorted by reach direction along both axes. **(b)** Example P-GPFA latent trajectories in the two principal latent dimensions for five rightward reaches (grey) and five leftward reaches (red). Trajectories are plotted from the appearance of the stimulus until movement onset (circles; the trajectories shown are the same as in Figure 3e). **(c)** As in (b), now showing latent trajectories inferred by factor analysis. These exhibit less discernable structure due to the lack of an explicit smoothness prior.

exhibited notable modulation at target onset (Figure 5e). In contrast, the neural data exhibited modulation by reach similarity 75 ms prior to movement onset. This modulation was strongest for the bGPFA latent states followed by the FA latents, and the modulation by reach similarity was very weak for the raw neural activity (Figure 5e). To see how this modulation by reach direction varied as a function of time from movement onset, we computed the difference ($\delta z$) between the maximum and minimum of the modulation curves and repeated this analysis at different times prior to and during the reach process. We found that the modulation in neural activity space increased before any detectable movement, with bGPFA showing the strongest signal followed by factor analysis and then the raw activity (Figure 5f). Indeed, the bGPFA latent modulation was maximized near movement onset, while the reach speed did not peak until several hundred milliseconds after movement onset where bGPFA latent trajectories have started to diverge again. Taken together, these results confirm that our analyses of bGPFA preparatory states do not reflect premature movement onset, and that they are not artifacts of the temporal correlations introduced by our GP prior since noisier but qualitatively similar results arise from the use of factor analysis.

For further comparison with non-Bayesian Gaussian process factor analysis, we also fitted Poisson GPFA (P-GPFA) to the primate dataset using our variational inference approach for scalability but without a prior over $C$ (Section 2.1). For this analysis, we used 16 latent dimensions as inferred by bGPFA, and we subselected latent processes with timescales $\leq 200$ ms to study the putatively fast motor preparation as for bGPFA. We then orthogonalized the latent dimensions by performing an SVD on the loading matrix (see 20 for details). Similar to our results from bGPFA, we found that the latent trajectories became modulated by movement direction prior to movement onset (Figure 6a) with a similar degree of modulation to bGPFA ($\delta z = 1.01$ for bGPFA; $\delta z = 0.99$ for P-GPFA). When visualizing the latent trajectories for the example reaches considered in Figure 3e, we also found that these diverged by reach direction (Figure 6b), similar to the bGPFA latent trajectories and unlike vanilla factor analysis which assumes temporal independence *a priori* (Figure 6c)

## B S1 activity

In this section, we compare the latent processes inferred for M1 dynamics to those inferred by applying bGPFA to the recordings from S1. In contrast to the clustering by reach direction in M1, there was less obvious modulation by movement direction in S1 prior to movement onset (Figure 7). To quantify this, we again computed the degree of modulation prior to movement onset, which was $\delta z = 0.38$ for S1 compared to $\delta z = 1.01$ for M1. This is also consistent with our decoding analyses in Figure 3b which showed that activity in M1 predicts movement 100-150 ms into the future while S1 activity is not predictive of future kinematics to the same extent.

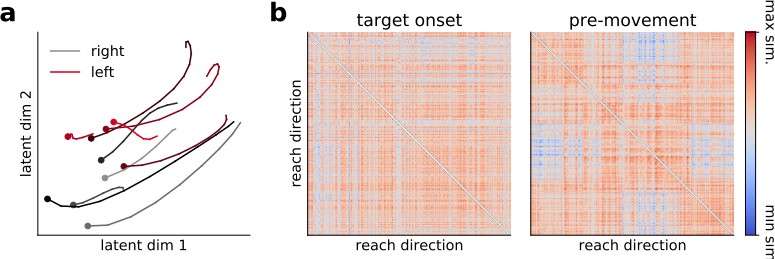

Figure 7: **Analyses of S1 dynamics by reach direction.** **(a)** bGPFA was fitted to data recorded from S1 during the self-paced reaching task. The panel shows example latent trajectories in the two most informative latent dimensions for five rightward reaches (grey) and five leftward reaches (red). Trajectories are plotted from the appearance of the stimulus until movement onset (circles; the trajectories shown are the same as in Figure 3e). Unlike the M1 latent trajectories, there is no obvious clustering by reach direction during movement preparation. **(b)** Similarity matrix of the latent state at stimulus onset (left) and 75 ms prior to movement onset (right). Reaches are sorted by reach direction along both axes, and there is no obvious structure in either similarity matrix in contrast to the results for the M1 recordings.

## C   Further reaction time analyses

For analyses of correlations between latent distances and reaction times, we only considered reaches with a reaction time of at least 125 ms and at most 425 ms, which retained 712 of 762 reaches (Figure 8a). This is because very long reaction times may reflect the monkey not being fully engaged with the task during those reaches, and very short reaction times may reflect spurious movement. To confirm that our finding of a strong correlation between latent distance and reaction time in Figure 3g is not an artifact of the temporal correlations introduced by the bGPFA generative model, we generated a synthetic control distribution. Here we drew 50,000 synthetic latent trajectories from our learned generative model with trajectory durations matched to those observed experimentally on each trial. We then computed mean preparatory states and latent distances to preparatory states as in the experimental data (Section 3.2) and computed correlations with the experimental reaction times. We found a mean correlation of $0.02$ and a range of $-0.14$ to $0.18$ in the synthetic data, suggesting that our generative model may introduce weak correlations between latent distances and reaction times. However, the experimentally observed correlation of $0.45$ was much larger than what could be expected by chance. This verifies our finding that the distance from the latent state at target onset to the corresponding preparatory state has behavioral relevance, with better initial states leading to shorter reaction times.

Although we already find a fairly strong relationship between latent distances and reaction times, it is worth noting that several additional considerations may further improve such predictions. Notably, our naïve Euclidean distance metric could be improved by instead defining a metric based on the probabilistic model itself [16]. Additionally, while we categorize reaches by reach direction, reaches in the same direction can still have different start and end points on the grid (Figure 3a), leading to different posture and muscle activations which is likely to significantly affect neural activity. Our analysis by reach direction therefore only represents a coarse categorization of the rich behavioral space, and it remains to be seen how neural activity and latent trajectories are affected by e.g. posture during the task.

Finally we considered how the dynamics of long-timescale latent processes relate to the reaction time across trials (c.f. Section 3.3). Here we found that the two slowest dimensions had timescales of $\tau = 1.4$ s and $\tau = 1.7$ s, similar to the timescale of single reaches which generally lasted between 1 and 2 seconds (Figure 8c). Intriguingly, the latent state in these dimensions at target onset was predictive of reaction time with Pearson correlations of $\rho = 0.40$ and $\rho = 0.36$ respectively (Figure 8d). While the information about reaction time contained in these two dimensions was largely redundant, it was orthogonal to that encoded by the distance to preparatory state in the fast dimensions. In particular, a linear model had 19.9% variance explained from the distance to prep in fast dimensions, 15.7% variance explained from the slow latent dimension with the strongest correlation, and 28.7% when combining these two features which corresponds to 80.7% of the additive value.

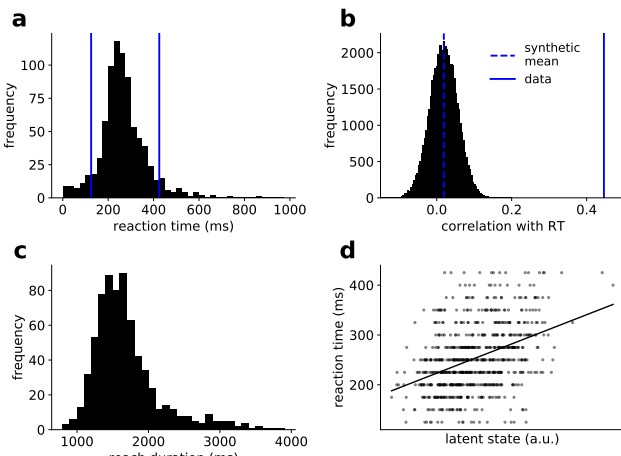

Figure 8: **Further reaction time analyses.** **(a)** Histogram of reaction times across all succesful reaches. For our correlation analyses, we only considered reaches with a reaction time between 125 ms and 425 ms (blue vertical lines). **(b)** Pearson correlations between distance to prep state and reaction time in synthetic data. Histogram corresponds to correlations between the true reaction times and 50,000 draws from the learned generative model. Blue dashed line indicates mean across all synthetic datasets (0.02), which is much smaller than the observed correlation in the experimental data of 0.45 (blue solid line). **(c)** Histogram of reach durations for all reaches with a reaction time between 125 ms and 425 ms. **(d)** Plot of reaction time against the value of a long timescale latent dimension at target onset ($\tau = 1.4$ s, $\rho = 0.40$).

## D   Latent dimensionality

In this section, we estimate the dimensionality of the primate data as a function of the offset between M1 and S1 spike times using both bGPFA and participation ratios computed on the basis of PCA [13]. The participation ratio is defined as

$$PR = \left( \sum_i \lambda_i \right)^2 / \sum_i \lambda_i^2, \tag{15}$$

where $\lambda_i$ is the $i^{th}$ eigenvalue of the covariance matrix $\boldsymbol{Y}\boldsymbol{Y}^T$. When computing the participation ratio of the data as a function of the M1 spike time shift, we find that the dimensionality is minimized for a shift of 75-100 ms (Figure 9). This suggests that the neural recordings can be explained more concisely when taking into account the offset in decoding between M1 and S1 which is consistent with the increased log likelihood after shifting the M1 spikes (Section 3.2).

This trend is not directly observable in the number of dimensions retained by bGPFA with and without a 100 ms shift of the M1 spike times ($18.8 \pm 0.19$ vs $18.7 \pm 0.20$ respectively across 10 model fits). However, the discrete nature of this dimensionality measure makes it relatively insensitive to small effects since it relies on stochastic differences in the retention of a dimension with little information content. We therefore utilized the interpretation of the learned prior scale $s_d^2$ as a measure of variance explained (Appendix J) and defined a 'participation ratio' for bGPFA, similar to the PCA participation ratio considered above:

$$PR_{bGPFA} = \left( \sum_d s_d^2 \right)^2 / \sum_d \left( s_d^2 \right)^2. \tag{16}$$

Here we found a strong effect of shifting the M1 spike times, which reduced the dimensionality from $PR_{bGPFA} = 6.16 \pm 0.12$ to $PR_{bGPFA} = 5.62 \pm 0.06$ ($p = 0.001$). Additionally, we note that bGPFA explains the data with only a handful of latent dimensions (19 total dimensions; 6 when re-weighted as a participation ratio). This is much lower than the dimensionality of 127-129 estimated by the PCA participation ratio which generally infers higher dimensionalities for noisier (more 'spherical') datasets.

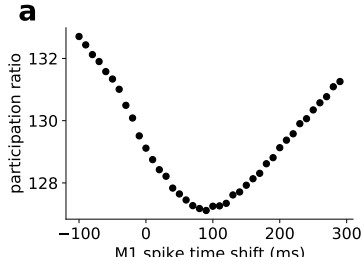

Figure 9: **Neural dimensionality. (a)** Participation ratio (Equation 15) as a function of temporal offset added to M1 spike times in the primate dataset.

## E    Further validation of bGPFA on synthetic and biological data

In Figure 2a-b, we considered the performance of FA, GPFA and bGPFA on synthetic data with Gaussian noise. To further validate our method in a non-Gaussian setting relevant to the study of electrophysiological recordings, we also performed similar analyses on (i) synthetic data with Poisson noise and (ii) experimental recordings from the primate reaching task. In both cases, we compared FA, GPFA and bGPFA with Poisson noise, since such Poisson noise models are common in the neuroscience literature [7, 11, 19, 21]. We note that these non-conjugate models are all readily implemented within our inference framework as special cases of bGPFA.

**Synthetic data**    In this section, we perform analyses similar to Figure 2b to validate bGPFA and its capacity for automatic relevance determination on synthetic data. We first generated a dataset drawn from the GPFA generative model but with a Poisson noise model after passing the activations through a softplus nonlinearity

$$\boldsymbol{Y} \sim Poisson\left(\log\left[1 + \exp \boldsymbol{C}\boldsymbol{X}\right]\right) \tag{17}$$

We then fitted Factor analysis, GPFA, and bGPFA and without ARD to the resulting dataset, all with Poisson observation models and exponential non-linearities (Appendix K; we denote these Poisson models as 'P-FA' etc.). To quantify performance, we computed the cross-validated predictive log likelihood $\mathcal{L}_{pred} = \sum_{it} \log p_{Poisson}(y_{it}|f_{it})$, where $i$ and $t$ index neurons and time points in a held-out test set (results were similar when considering MSEs). When considering $\mathcal{L}_{pred}$ as a function of latent dimensionality, we found that both P-FA and P-GPFA exhibited a clear maximum at the true dimensionality of $D^* = 3$ while P-bGPFA without ARD was robust to overfitting, similar to our findings for the Gaussian models (Figure 2b). Finally, bGPFA with ARD was capable of automatically recovering this dimensionality as well as the maximum predictive performance achieved across the other models.

**Experimental data**    We then proceeded to perform an analysis as above on data from the self-paced monkey reaching dataset [9]. For this analysis, we used a smaller subset of the data for computational convenience, considering only 1000 timepoints but including all 200 neurons. We performed these analyses in 10-fold cross-validation, averaging performance over folds and repeating the entire analysis across 5 different random seeds. We again fitted P-FA and P-GPFA and found that these models exhibited a clear maximum in their predictive log likelihoods. As for the synthetic data, P-bGPFA without ARD was robust to overfitting, and the introduction of ARD allowed us to infer the optimal latent dimensionality of $D^* \approx 4$ as well as achieving optimal performance without *a priori* assumptions about the latent dimensionality. The robustness to overfitting of bGPFA both with and without ARD suggests that it could also be a valuable tool in settings with large simultaneous recordings of thousands of neurons, which are becoming increasingly relevant with recent advances in neural recording technologies [10, 15].

Taken together, these results further validate the utility of bGPFA on both synthetic and biological data with non-conjugate noise models. They also highlight the utility of automatic relevance determination in practice, where it obviates the need to perform extensive cross-validation to select an appropriate latent dimensionality for the experimental data.

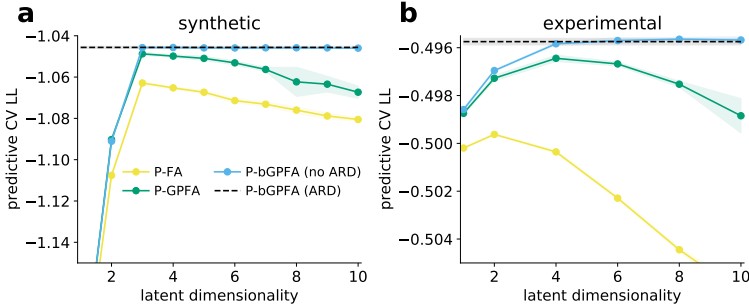

Figure 10: **Bayesian GPFA applied to spike count data.** **(a)** Cross-validated predictive log likelihoods of factor analysis (yellow), GPFA (green), and Bayesian GPFA without ARD (blue) fitted to synthetic data with a ground truth dimensionality of three for different model dimensionalities. All methods used a Poisson observation model $p(y_{nt}|f_{nt})$. bGPFA with ARD recovered the performance of the optimal non-ARD models without requiring a search over latent dimensionalities (black). **(b)** As in (a), now for models applied to a subset of the monkey reaching data analyzed in Section 3.2.

## F Parameterizations of the approximate GP posterior

In this section, we compare different forms of the variational posterior $q(\boldsymbol{X})$ discussed in Section 2.2. For factorizing likelihoods, the optimal posterior takes the form

$$q(\boldsymbol{x}_d) \propto p(\boldsymbol{x}) \prod_t \mathcal{N}(x_t|g_t, v_t), \tag{18}$$

where $g_t$ and $v_t$ are variational parameters [8]. Equation 18 might therefore seem to be an appropriate form of the variational distribution $q(\boldsymbol{X})$. However, this formulation is computationally expensive and the likelihood $p(\boldsymbol{Y}|\boldsymbol{X})$ does not factorize across time in bGPFA.

Instead, we therefore consider approximate parameterizations of the form

$$q(\boldsymbol{x}_d) = \mathcal{N}(\boldsymbol{\mu}_d, \boldsymbol{\Sigma}_d) \tag{19}$$

$$\boldsymbol{\mu}_d = \boldsymbol{K}_d^{\frac{1}{2}} \boldsymbol{\nu}_d \tag{20}$$

$$\boldsymbol{\Sigma}_d = \boldsymbol{K}_d^{\frac{1}{2}} \boldsymbol{\Lambda}_d \boldsymbol{\Lambda}_d^T \boldsymbol{K}_d^{\frac{1}{2}}, \tag{21}$$

where $\boldsymbol{K}_d^{\frac{1}{2}}$ is a matrix square root of the prior covariance matrix $\boldsymbol{K}_d$, and $\boldsymbol{\nu}_d \in \mathbb{R}^T$ is a vector of variational parameters. This formulation simplifies the KL divergence term for each latent dimension in Equation 6 from

$$\text{KL}[q(\boldsymbol{x}_d)||p(\boldsymbol{x}_d|\boldsymbol{t})] = \frac{1}{2}\left(\text{Tr}(\boldsymbol{K}_d^{-1}\boldsymbol{\Sigma}_d) + \log|\boldsymbol{K}_d| - \log|\boldsymbol{\Sigma}_d| + \boldsymbol{\mu}_d^T\boldsymbol{K}_d^{-1}\boldsymbol{\mu}_d - T\right) \tag{22}$$

to

$$\text{KL}[q(\boldsymbol{x}_d)||p(\boldsymbol{x}_d|\boldsymbol{t})] = \frac{1}{2}\left(||\boldsymbol{\Lambda}_d||_\text{F}^2 - 2\log|\boldsymbol{\Lambda}_d| + ||\boldsymbol{\nu}_d||^2 - T\right). \tag{23}$$

In the following, we drop the $\cdot_d$ subscript to remove clutter, and we use the notation $\boldsymbol{\Psi} = \text{diag}(\psi_1, ..., \psi_T)$ with positive elements $\psi_t > 0$, to denote a positive definite diagonal matrix.

### F.1 Square root of the prior covariance

For a stationary prior covariance $\boldsymbol{K}$, we can directly parameterize $\boldsymbol{K}^{\frac{1}{2}}$ by taking the square root of $k(\cdot, \cdot)$ in the Fourier domain and computing the inverse Fourier transform. For the RBF kernel used in this work we get

$$k(t_i, t_j) = \exp\left(-\frac{(t_i - t_j)^2}{2\tau^2}\right) \tag{24}$$

$$k^{\frac{1}{2}}(t_i, t_j) = \left(\frac{2}{\pi}\right)^{\frac{1}{4}} \left(\frac{\delta t}{\tau}\right)^{\frac{1}{2}} \exp\left(-\frac{(t_i - t_j)^2}{\tau^2}\right). \tag{25}$$

In this expression, $\delta t$ is the time difference between consecutive data points, we have assumed a signal variance of 1 in the prior kernel, and we note that our parameterization only gives rise to the exact matrix square root of the RBF kernel in the limit where $T \gg \tau$. Note that this is the case in the present work since $T \approx 30$ minutes is much larger than the longest timescales learned by bGPFA ($\tau \approx 2$ s). For most experiments in neuroscience, observations are binned such that time is on a regularly spaced grid and our parameterization can be applied directly. In other cases, kernel interpolation should first be used to construct a covariance matrix with Toeplitz structure [17, 18].

### F.2 Parameterization of the posterior covariance

We now proceed to describe the various parameterizations of $\boldsymbol{\Lambda}$ whose performance is compared in Figure 11. Other parameterizations are explored in [2].

**Diagonal $\boldsymbol{\Lambda}$**   We parameterize each latent dimension with $\boldsymbol{\Lambda} = \boldsymbol{\Psi}$. This gives rise to a KL term:

$$2\text{KL}[q(\boldsymbol{x})||p(\boldsymbol{x})] = \sum_t \psi_t^2 + ||\boldsymbol{\nu}||^2 - T - 2\sum_t \log \psi_t. \tag{26}$$

We can compute $\boldsymbol{\Lambda}\boldsymbol{v}$ in linear time since $\boldsymbol{\Lambda}$ is diagonal which allows for cheap differentiable sampling:

$$\boldsymbol{\eta} \sim \mathcal{N}(0, \boldsymbol{I}) \tag{27}$$

$$\text{sample} = \boldsymbol{K}^{\frac{1}{2}}(\boldsymbol{\Lambda}\boldsymbol{\eta} + \boldsymbol{\nu}), \tag{28}$$

where the multiplication by $\boldsymbol{K}^{\frac{1}{2}}$ is done in $\mathcal{O}(T \log T)$ time in the Fourier domain.

**Circulant $\boldsymbol{\Lambda}$**   We parameterize each latent dimension with $\boldsymbol{\Lambda} = \boldsymbol{\Psi}\boldsymbol{C}$. Here, $\boldsymbol{C} \in \mathbb{R}^{T \times T}$ is a positive definite circulant matrix with $1 + \frac{T}{2}$ (integer division) free parameters, which we parameterize directly in the Fourier domain as $\hat{\boldsymbol{c}} = \text{rfft}(\boldsymbol{c}) \in \mathbb{R}^{1 + T/2}$, where $\boldsymbol{c}$ is the first column of $\boldsymbol{C}$ with $\hat{\boldsymbol{c}} \geq 0$ elementwise. We compute the KL as

$$2\text{KL}[q(\boldsymbol{x})||p(\boldsymbol{x})] = \left(\sum_t c_t^2\right)\left(\sum_t \psi_t^2\right) + ||\boldsymbol{\nu}||^2 - T - 2\sum_t \log \psi_t - 2\log|\boldsymbol{C}| \tag{29}$$

$$\log|\boldsymbol{C}| = \log \hat{c}_1 + \log \hat{c}_{\frac{T}{2}+1} + 2\sum_{i=2}^{\frac{T}{2}} \log \hat{c}_i \qquad \text{(even T)} \tag{30}$$

$$\log|\boldsymbol{C}| = \log \hat{c}_1 + 2\sum_{i=2}^{\frac{T+1}{2}} \log \hat{c}_i \qquad \text{(odd T)}, \tag{31}$$

where $\boldsymbol{c} = \text{irfft}(\hat{\boldsymbol{c}})$. We can sample differentiably in $\mathcal{O}(T \log T)$ time by computing

$$\boldsymbol{\eta} \sim \mathcal{N}(0, \boldsymbol{I}) \tag{32}$$

$$\boldsymbol{C}\boldsymbol{\eta} = \text{irfft}(\hat{\boldsymbol{c}} \odot \text{rfft}(\boldsymbol{\eta})) \tag{33}$$

$$\text{sample} = \boldsymbol{K}^{\frac{1}{2}}(\boldsymbol{\Psi}\boldsymbol{C}\boldsymbol{\eta} + \boldsymbol{\nu}), \tag{34}$$

where $\odot$ denotes the complex element-wise product.

**Low-rank $\boldsymbol{\Lambda}$**   We let $\boldsymbol{Q} \in V_r(\mathbb{R}^T)$ such that $\boldsymbol{Q}^T\boldsymbol{Q} = \boldsymbol{I}_r$ and write

$$\boldsymbol{\Lambda} = \boldsymbol{I}_T - \boldsymbol{Q}\boldsymbol{\Psi}\boldsymbol{Q}^T, \tag{35}$$

where we now constrain $0 < \psi_i < 1$ to maintain the positive definiteness of $\boldsymbol{\Lambda}$. In practice, we keep $\boldsymbol{Q}$ on the Stiefel manifold (i.e. $\boldsymbol{Q}^T\boldsymbol{Q} = \boldsymbol{I}_r$) by (differentiably) computing the QR decomposition of a $T \times r$ matrix of free parameters.

**Circulant inverse $\boldsymbol{\Lambda}$**   We let $\boldsymbol{C}$ be a circulant positive definite matrix as above and parameterize

$$\boldsymbol{\Lambda} = (\boldsymbol{I} + \boldsymbol{\Psi}\boldsymbol{C}\boldsymbol{\Psi})^{-1}. \tag{36}$$

Computing $\boldsymbol{\Lambda}\boldsymbol{v}$ products is done using the conjugate gradient algorithm, taking advantage of fast products with $\boldsymbol{\Psi}$ and $\boldsymbol{C}$; the same algorithm is also used to stochastically estimate $\log|\boldsymbol{\Lambda}|$ and its gradient (see the appendix of 14).

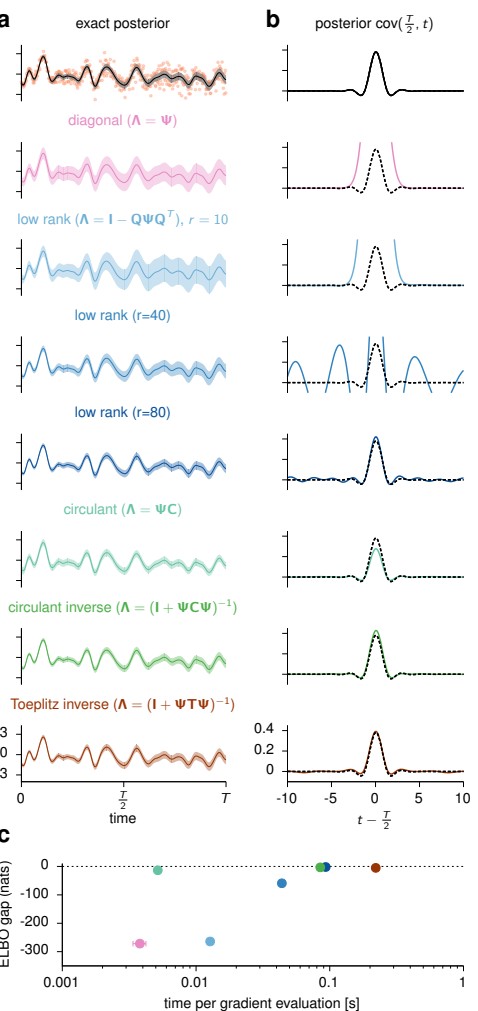

Figure 11: **Comparisons of different forms of the approximate posterior** $q(\boldsymbol{x})$. **(a)** Synthetic data (orange dots) plotted together with the exact posterior (black) as well as the variational posteriors inferred by each whitened parameterization. The solid lines denote the (approximate) posterior means, and shaded areas indicate $\pm 1$ posterior standard deviations. **(b)** Slice through the posterior covariance $(\mathrm{Cov}_{x \sim q(x)}\left[x_{T/2}, x_t\right])$ for the true posterior (top and black dotted lines) and the approximate methods. Each method has different characteristics, and the circulant parameterization provides a good qualitative fit at very low computational cost. **(c)** We defined the 'ELBO gap' of each method as ELBO $-$ LL, where LL is the true data log likelihood. We plotted this against the time per gradient evaluation and found that the circulant parameterization achieved high accuracy with cheap gradients.

**Toeplitz inverse $\boldsymbol{\Lambda}$**   This proceeds just as for the circulant inverse form, with the circulant matrix $\boldsymbol{C}$ replaced by an arbitrary Toeplitz matrix (also exploiting fast $\boldsymbol{Tv}$ products):

$$\boldsymbol{\Lambda} = (\boldsymbol{I} + \boldsymbol{\Psi T \Psi})^{-1}. \tag{37}$$

### F.3   Numerical comparisons between different parameterizations

To compare these parameterizations, we generated a synthetic dataset (Figure 11a, orange dots) over $T = 1000$ time bins by drawing samples $\{y_1, \ldots, y_T\}$ as $y_t = x_t + \sigma_t \xi_t$ where $\xi(t) \sim \mathcal{N}(0, 1)$ with non-stationary $\sigma_t$ growing linearly from 0.1 to 0.5 over the whole range $0 \leq t < T$, and $x_i \sim \mathcal{N}(0, \boldsymbol{K}^{1/2}\boldsymbol{K}^{1/2})$ with $\boldsymbol{K}^{1/2}$ given by Equation 25. We fixed these generative parameters to their ground truth and optimized the ELBO w.r.t. the variational parameters in this simple regression setting. We found that all of the parameterizations accurately recapitulated the GP posterior mean (Figure 11a). However, the degree to which they captured the non-stationary posterior covariance and data log likelihood varied between methods (Figure 11b-c). To quantify this, we computed the difference between the asymptotic ELBO of each method and the exact log marginal likelihood. This ELBO gap was small for the circulant parameterization, the inverse methods, and the low rank parameterization with sufficiently high $r$. Although the circulant parameterization did not fully capture the non-stationary aspect of the posterior variance, this did not affect the ELBO gap substantially. Importantly, however, the circulant parameterization was more than an order of magnitude faster per gradient evaluation than the other methods with comparable accuracy (Figure 11c). For these reasons as well as the excellent performance in a latent variable setting (Section 3.1, Section 3.2, Appendix E), we used the circulant parameterization for all experiments. However, repeating all

analyses with a simple diagonal parameterization also lead to good performance and qualitatively similar results.

# G   Relation between variational posterior over $F$ and true posterior

Here we show that our parameterization of $q(\boldsymbol{f}_n)$ includes the exact posterior in the case of Gaussian noise.

When the noise model is Gaussian (i.e., $p(\boldsymbol{y}_n|\boldsymbol{f}_n) = \mathcal{N}(\boldsymbol{y}|\boldsymbol{f}_n, \sigma_n^2 I)$), we can compute the posterior over $\boldsymbol{f}_n^* = f_n(\boldsymbol{X}^\star)$ at locations $\boldsymbol{X}^\star$ in closed form:

$$\boldsymbol{f}_n^*|\boldsymbol{X}^\star, \boldsymbol{X}, \boldsymbol{y}_n \sim \mathcal{N}(\boldsymbol{X}^{\star T}\boldsymbol{S}^2\boldsymbol{X}\hat{\boldsymbol{K}}^{-1}\boldsymbol{y}_n, \boldsymbol{X}^{\star T}\boldsymbol{S}(\boldsymbol{I} - \boldsymbol{X}\hat{\boldsymbol{K}}^{-1}\boldsymbol{X}^T)\boldsymbol{S}\boldsymbol{X}^\star) \tag{38}$$

where $\hat{\boldsymbol{K}} = \boldsymbol{X}^T\boldsymbol{S}^2\boldsymbol{X} + \sigma_n^2\boldsymbol{I}$. Note that the posterior is low-rank as the rank of $\boldsymbol{I} - \boldsymbol{X}\hat{\boldsymbol{K}}^{-1}\boldsymbol{X}^T$ is at most $D$. This means that when we do variational inference, we can parameterize our approximate posterior as:

$$q(\boldsymbol{f}_n^*) = \mathcal{N}(\boldsymbol{f}|\boldsymbol{X}^{\star T}\boldsymbol{S}\boldsymbol{\nu}_n, \boldsymbol{X}^{\star T}\boldsymbol{S}\boldsymbol{L}_n\boldsymbol{L}_n^T\boldsymbol{S}\boldsymbol{X}^\star) \tag{39}$$

where $\boldsymbol{\nu}_n \in \mathbb{R}^D$ and $\boldsymbol{L}_n \in \mathbb{R}^{D \times D}$ are the parameters of the approximate posterior (Section 2.2). We see that this parameterization is exact when:

$$\boldsymbol{\nu}_n = \boldsymbol{S}\boldsymbol{X}\hat{\boldsymbol{K}}^{-1}\boldsymbol{y}_n \tag{40}$$

$$\boldsymbol{L}_n\boldsymbol{L}_n^T = \boldsymbol{I} - \boldsymbol{X}\hat{\boldsymbol{K}}^{-1}\boldsymbol{X}^T. \tag{41}$$

Note that the right-hand side of Equation 41 is guaranteed to be positive definite because the true posterior must be positive definite. Importantly, for this parameterization, the KL term in Equation 10 simplifies to

$$\mathrm{KL}(q(\boldsymbol{f}_n|\boldsymbol{X})||p(\boldsymbol{f}_n|\boldsymbol{X})) = \mathrm{KL}(\mathcal{N}(\boldsymbol{\nu}_n, \boldsymbol{L}_n\boldsymbol{L}_n^T)||\mathcal{N}(0, \boldsymbol{I})), \tag{42}$$

which is independent of $\boldsymbol{X}$ and allows us to do efficient inference due to the low dimensionality of $\boldsymbol{\nu}_n$ and $\boldsymbol{L}_n$.

# H   Relation between variational posterior over $F$ and SVGP

For general non-Gaussian noise models, the parameterization in Appendix G will no longer be exact. However, here we show that it is in this case equivalent to a stochastic variational Gaussian process (SVGP; 3). In SVGP, we choose a variational distribution:

$$q(\boldsymbol{u}) = \mathcal{N}(\boldsymbol{u}|\boldsymbol{Z}^T\boldsymbol{S}\boldsymbol{\mu}, \boldsymbol{Z}^T\boldsymbol{S}\boldsymbol{M}\boldsymbol{M}^T\boldsymbol{S}\boldsymbol{Z}) \tag{43}$$

at inducing points $\boldsymbol{Z} \in \mathbb{R}^{D \times m}$, where $\boldsymbol{\mu}$ and $\boldsymbol{M}$ are the "whitened" parameters [4]. This gives an approximate posterior:

$$q(\boldsymbol{f}^*) = \mathbb{E}_{q(\boldsymbol{u})}\left[p(\boldsymbol{f}|\boldsymbol{u})\right] \tag{44}$$

$$= \mathcal{N}(\boldsymbol{f}|\boldsymbol{X}^{\star T}\boldsymbol{S}\boldsymbol{\Pi}_z\boldsymbol{\mu}; \boldsymbol{X}^{\star T}\boldsymbol{S}\boldsymbol{\Pi}_z(\boldsymbol{M}\boldsymbol{M}^T - \boldsymbol{I})\boldsymbol{\Pi}_z\boldsymbol{S}\boldsymbol{X}^\star) \tag{45}$$

where $\boldsymbol{\Pi}_z = \boldsymbol{S}\boldsymbol{Z}(\boldsymbol{Z}^T\boldsymbol{S}^2\boldsymbol{Z})^{-1}\boldsymbol{Z}^T\boldsymbol{S}$. If we choose $m = D$ inducing points such that $\boldsymbol{Z} \in \mathbb{R}^{D \times D}$ and make sure $\boldsymbol{Z}$ has full rank, then $\boldsymbol{\Pi}_z = \boldsymbol{I}$ and thus

$$q(\boldsymbol{f}^*) = \mathcal{N}(\boldsymbol{f}|\boldsymbol{X}^{\star T}\boldsymbol{S}\boldsymbol{\mu}, \boldsymbol{X}^{\star T}\boldsymbol{S}(\boldsymbol{M}\boldsymbol{M}^T - \boldsymbol{I})\boldsymbol{S}\boldsymbol{X}^\star). \tag{46}$$

We recover the parameterization in Section 2.2 when

$$\boldsymbol{\mu} = \boldsymbol{\nu} \quad \text{and} \quad \boldsymbol{M}\boldsymbol{M}^T - \boldsymbol{I} = \boldsymbol{L}\boldsymbol{L}^T. \tag{47}$$

For these more general noise models, the whitened parameterization of $q(\boldsymbol{f})$ still gives rise to a computationally cheap KL divergence that is independent of $\boldsymbol{X}$ as in Equation 42:

$$\mathrm{KL}(q(\boldsymbol{f}_n|\boldsymbol{X})||p(\boldsymbol{f}_n|\boldsymbol{X})) = \mathrm{KL}(\mathcal{N}(\boldsymbol{\nu}_n, \boldsymbol{L}_n\boldsymbol{L}_n^T)||\mathcal{N}(0, \boldsymbol{I})). \tag{48}$$

In summary, we have shown that (i) our parameterization of $q(\boldsymbol{f}_n)$ has sufficient flexibility to learn the true posterior when the noise model is Gaussian (Appendix G), and (ii) it is equivalent to performing SVGP where the locations of the inducing points do not matter provided that their rank is at least as high as the number of latent dimensions.

# I  Automatic relevance determination

Here we briefly consider why introducing a prior over the factor matrix enables automatic relevance determination. These ideas reflect results by Bishop [1] and our experiments in Section 3.1.

For simplicity, we will first consider the case of factor analysis where $p(\boldsymbol{X}) = \prod_{d,t} \mathcal{N}(x_{dt}; 0, 1)$. This gives rise to a marginal likelihood (with Gaussian noise) equal to

$$\log p(\boldsymbol{Y}) = \sum_t \log \mathcal{N}(\boldsymbol{y}_t; 0, \boldsymbol{C}\boldsymbol{C}^T + \boldsymbol{\Sigma}), \tag{49}$$

where $\boldsymbol{\Sigma} = \mathrm{diag}(\sigma_1^2, ..., \sigma_N^2)$ is a diagonal matrix of noise parameters. It is in this case quite clear that the optimal marginal likelihood is a monotonically increasing function of the latent dimensionality, since any marginal likelihood reachable with a certain rank $D$ is also reachable with a larger rank $D' > D$; increasing $D$ can only increase model flexibility. We could in this case threshold the magnitude of the columns of $\boldsymbol{C}$ to subselect more 'informative' dimensions, but this is not inherently different from putting an arbitrary cut-off on the variance explained in PCA, and there is no Bayesian "Occam's razor" built into the method [6].

Consider now the case where we put a unit Gaussian prior on $c_{nd}$. In this case, $\{c_{nd}\}$ are no longer parameters of the model but rather latent variables to be inferred, which intuitively should reduce the risk of overfitting. To expand on this intuition, consider the ELBO (c.f. Section 2.1) that results from introducing such a prior over $c_{nd}$:

$$\log p(\boldsymbol{Y}) \geq \mathbb{E}_{q(\boldsymbol{X})}\left[\log p(\boldsymbol{Y}|\boldsymbol{X})\right] - \sum_{d,t} \mathrm{KL}\left[q(x_{dt})||\mathcal{N}(0,1)\right] \tag{50}$$

$$\log p(\boldsymbol{Y}|\boldsymbol{X}) = \sum_n \log \mathcal{N}(\boldsymbol{y}_n; 0, \boldsymbol{X}^T\boldsymbol{X} + \sigma_n \boldsymbol{I}). \tag{51}$$

Here we see that if a dimension $d$ is truly uninformative, it should have $x_{dt} = 0 \ \forall_t$ to avoid contributing noise to the likelihood term via $\boldsymbol{X}^T\boldsymbol{X}$. However, reducing this noise will increase the prior KL term, driving it to infinity in the limit of zero noise since the variational posterior over the $d^{th}$ latent at time $t$, $q(x_{dt})$, is in this case a delta function at zero. Optimizing the ELBO therefore involves a balance between mitigating the noise induced by $\boldsymbol{X}^T\boldsymbol{X}$ and reducing the KL penalty, with both of these terms contributing to a decreased ELBO compared to the model without uninformative dimensions. Thus the prior over $c_{nd}$ counteracts the overfitting that would normally occur when increasing the latent dimensionality in classical factor analysis, and this Bayesian treatment will lead to a decrease in the ELBO with increasing dimensionality beyond the optimal $D^\star$ that is needed to adequately explain the data.

Finally let us consider the case where we learn the prior scale of the factor matrix, such that $c_{nd} \sim \mathcal{N}(0, s_d^2)$ with $s_d$ optimized w.r.t. the ELBO. Critically, the likelihood term now becomes:

$$\log p(\boldsymbol{Y}|\boldsymbol{X}) = \sum_n \log \mathcal{N}(\boldsymbol{y}_n; 0, \boldsymbol{X}^T\boldsymbol{S}^2\boldsymbol{X} + \sigma_n \boldsymbol{I}). \tag{52}$$

with $\boldsymbol{S} = \mathrm{diag}(s_1, \ldots, s_D)$. In this case, adding uninformative dimensions beyond the optimal $D^\star$ still cannot increase the ELBO (in the limit of large $N$). However, letting $s_d \to 0$ for these superfluous dimensions will prevent them from contributing to $p(\boldsymbol{Y}|\boldsymbol{X})$, thus allowing $q(x_{dt}) \to \mathcal{N}(0,1)$ to drive the prior KL term to zero for these dimensions. In this limit, we recover both the ELBO and the posteriors associated with the $D^\star$- dimensional model. We thus have a built-in Occam's razor which will shave off any uninformative latent dimensions, and these will be identifiable as dimensions for which $s_d \approx 0$ and $q(x_{dt}) \approx \mathcal{N}(0,1)$.

These ideas generalize to GPFA where the posterior over latents will instead approach the GP prior $q(\boldsymbol{x}_d) \approx \mathcal{N}(0, \boldsymbol{K})$ for uninformative dimensions. This corresponds to the limit of $\boldsymbol{\nu} \to 0$, $\boldsymbol{C} \to \boldsymbol{I}$, and $\boldsymbol{\Psi} \to \boldsymbol{I}$ in our circulant parameterization in Section 2.2 and Appendix F. In all of our simulations, we found a clear clustering of dimensions after training with some clustered near zero $s_d$, and others clustered with much larger $s_d$ (Figure 2c and Figure 3b). Note that in practice we do not actively truncate the model by discarding dimensions with $s_d \approx 0$ but merely use the terminology to indicate that these dimensions have negligible contributions to the posterior predictive $q(\boldsymbol{y}_n)$, as well as to the latent posteriors $q(\boldsymbol{x}_d)$ for the dimensions with large $s_d$.

## J  Most informative dimensions

In this work, we refer to the latent dimensions with the highest values of $s_d$ as the 'most informative dimensions'. We do this because (i) observing the value of the corresponding latent $x_d$ decreases the variance of the expected distribution of neural activity more as $s_d$ increases, and (ii) the Fisher information of $x_d$ increases as $s_d$ increases.

To show this, we consider how the distribution over $f_n$ (the activity of neuron $n$) given $\mathbf{c}_n$ (the $n^{\text{th}}$ row of $\mathbf{C}$) changes when $x_d$ (the value of the $d^{th}$ latent) is known, and how this varies with $s_d$. In the following, we omit the $\cdot_n$ subscript for notational simplicity, and we note that $f$, $x_d$ and $c_d$ are all scalar values. With unknown $x_d$, $f$ is Gaussian with zero mean and variance $\mathbb{E}_{p(\mathbf{x})}\left[\mathbf{c}^T \mathbf{x} \mathbf{x}^T \mathbf{c}\right] = \mathbf{c}^T \mathbf{c}$. Thus,

$$p(f|\mathbf{c}) = \mathcal{N}(f; 0, \mathbf{c}^T \mathbf{c}) \tag{53}$$

In contrast, for known $x_d$, we have

$$p(f|\mathbf{c}, x_d) = \mathcal{N}(f; c_d x_d, \mathbf{c}_{-d}^T \mathbf{c}_{-d}), \tag{54}$$

where $\mathbf{c}_{-d}$ is $\mathbf{c}$ with the $d^{\text{th}}$ element removed. We thus see that the decrease in variance of $f$ from observing $x_d$ is $c_d^2$. Finally, we can approximate the process of averaging this quantity over neurons by noting that $c_d \sim \mathcal{N}(0, s_d^2)$ and marginalising out $\mathbf{c}$:

$$\mathbb{E}_{p(\mathbf{c})}[\sigma_{f|\mathbf{c}}^2 - \sigma_{f|\mathbf{c}, x_d}^2] = \mathbb{E}_{p(\mathbf{c})}[c_d^2] = s_d^2, \tag{55}$$

where $\sigma_{f|\mathbf{c}}^2$ is the variance of $p(f|\mathbf{c})$. Thus, $s_d^2$ can be interpreted as the expected decrease in the variance of the denoised neural activity $f$ when learning the value of the $d^{th}$ latent.

This can also be understood in information-theoretic terms by considering the Fisher information of the $d^{th}$ latent dimension which is given by

$$\mathcal{I}(x_d|\mathbf{c}) = -\mathbb{E}_{p(f|x_d, \mathbf{c})}\left[\frac{\partial^2}{\partial x_d^2} \log p(f|x_d, \mathbf{c})\right] \tag{56}$$

$$= \left[\sum_{d' \neq d} c_{d'}^2\right]^{-1}. \tag{57}$$

To relate this quantity to our prior scale parameters $\{s_d\}$, we consider the expectation of the inverse Fisher information:

$$\mathbb{E}_{p(\mathbf{c})}[\mathcal{I}(x_d|\mathbf{c})^{-1}] = \sum_{d' \neq d} s_{d'}^2. \tag{58}$$

For a given set of latent dimensions $[1, D]$ with corresponding $\{s_d\}_1^D$, we thus see that the expected *inverse* Fisher information is *minimized* for the dimension with the highest value of $s_d$. In Figure 2 and Figure 3 we use $s_d$ together with the posterior latent mean parameters $\boldsymbol{\nu}_d$ to identify 'discarded' dimensions.

## K  Noise models and evaluation of their expectations

**Gaussian**   The Gaussian noise model is given by

$$\log p(y_{nt}|f_{nt}) = -\frac{1}{2}\log(2\pi) - \frac{1}{2}(y_{nt} - f_{nt})^2/\sigma_n^2, \tag{59}$$

where $\sigma_n$ is a learnable parameter. In this case we can easily compute the expected log-density under the approximate posterior analytically:

$$\mathbb{E}_{q(f_{nt}|\mathbf{X})}[\log p(y_{nt}|f_{nt})] = -\frac{1}{2}\left(\log(2\pi) + \frac{(y_{nt} - \mu_{nt})^2 + \Sigma_{ntt}}{\sigma_n^2}\right), \tag{60}$$

where $q(\mathbf{f}_n|\mathbf{X}) = \mathcal{N}(\mathbf{f}_n; \boldsymbol{\mu}_n, \boldsymbol{\Sigma}_n)$ and $\Sigma_{ntt}$ is the approximate posterior variance of neuron $n$ at time $t$ (i.e., the $t^{\text{th}}$ diagonal element of $\boldsymbol{\Sigma}_n$).

**Poisson**   The Poisson noise model is given by

$$\log p(y_{nt}|f_{nt}) = y_{nt} \log g(f_{nt}) - g(f_{nt}) - \log(y_{nt}!), \tag{61}$$

where $g$ is a link function. If we choose an exponential link function (i.e., $g(x) = \exp(x)$), we can compute in closed-form the expected log-density of the approximate posterior as:

$$\mathbb{E}_{q(f_{nt}|\boldsymbol{X})}\left[\log p(y_{nt}|f_{nt})\right] = \mathbb{E}_{q(f_{nt}|\boldsymbol{X})}\left[y_{nt}f_{nt} - \exp(f_{nt}) - \log(y_{nt}!)\right] \tag{62}$$

$$= y_{nt}\mu_{nt} - \exp\left(\mu_{nt} + \frac{1}{2}\Sigma_{ntt}\right) - \log(y_{nt}!). \tag{63}$$

For the analyses shown in Figure 2c-d, we use the exponential link function.

For general link functions $g$, we may not be able to evaluate the expected log-density in closed-form. In this case, we approximate it with Gauss-Hermite quadrature:

$$\mathbb{E}_{q(f_{nt}|\boldsymbol{X})}\left[\log p(y_{nt}|f_{nt})\right] \approx \frac{1}{\sqrt{\pi}} \sum_{i=1}^{k_{\mathrm{GH}}} \omega_i \log p(y_{nt}|f_{nt}^{(i)}) \tag{64}$$

where

$$\omega_i = \frac{2^{k_{\mathrm{GH}}-1} k_{\mathrm{GH}}! \sqrt{\pi}}{k_{\mathrm{GH}}^2 [H_{k_{\mathrm{GH}}-1}(r_i)]^2}, \tag{65}$$

$$f_{nt}^{(i)} = \left(\sqrt{2\Sigma_{ntt}}\right) r_i + \mu_{nt}, \tag{66}$$

$H_k(r)$ are the physicist's Hermite polynomials, and $r_i$ with $i = 1, \ldots, k$ are roots of $H_k(r)$. For a given order of approximation $k_{\mathrm{GH}}$, we can evaluate both $\omega_i$ and $r_i$ using standard numerical software packages such as Numpy. In practice, we find that $k_{\mathrm{GH}} = 20$ gives an accurate approximation to the expected log-density under the approximate posterior. Note that we could also estimate the expectation over $q(f_{nt})$ for general link functions $g$ using a Monte Carlo estimate, but we use Gauss-Hermite quadrature in this work since it has a lower computational cost and lower variance.

**Negative binomial**   The negative binomial noise model is given by

$$\log p(y_{nt}|f_{nt}) = \log \binom{y_{nt} + \kappa_n - 1}{y_{nt}} + \kappa_n \log\left(1 - g(f_{nt})\right) + y \log\left(g(f_{nt})\right), \tag{67}$$

where $g(f_{nt})$ denotes the probability of success in a Bernoulli trial. Here, each success corresponds to the emission of one spike in bin $t$, and thus $p(y_{nt}|f_{nt})$ is the distribution over the number of successful trials (spikes) before reaching $\kappa_n$ failed trials. The link function $g(x) : \mathbb{R} \to [0, 1)$ maps $f_{nt}$ to a real number between 0 and 1. In practice we use a sigmoid link-function $g(x) = 1/(1 + \exp(-x))$.

In this model, $\kappa_n$ is a learnable parameter which effectively modulates the overdispersion of the distribution since the mean and variance of $p(y_{nt}|f_{nt})$ are given by:

$$\mu_{NB} = \frac{g(f_{nt})\kappa_n}{1 - g(f_{nt})} \tag{68}$$

$$\sigma_{NB}^2 = \mu_{NB}\left(1 + \frac{\mu_{NB}}{\kappa_n}\right). \tag{69}$$

This is the parameter which we compare between the ground truth and trained models in Figure 2, and we see that the Poisson model is recovered for neuron $n$ as $\kappa_n \to \infty$.

For the negative binomial noise model we cannot compute the expected log-density in closed-form. We instead approximate this expectation using Gauss-Hermite quadrature as described above.

## L   Implementation

In this section, we provide pseudocode for bGPFA (Algorithm 1) with the circulant parameterization for $q(\boldsymbol{X})$ and discuss other implementation details.

**Algorithm 1:** Bayesian GPFA with automatic relevance determination

---

1  **input:** data $\boldsymbol{Y} \in \mathbb{R}^{N \times T}$, maximum latent dimensionality $D$, # of Monte Carlo samples $M$, learning rate $\gamma$

2  **parameters:** $\theta = \{\{s_d\}_1^D, \{\tau_d\}_1^D, \{\boldsymbol{\nu}_d\}_1^D, \{\tilde{\boldsymbol{c}}_d\}_1^D, \{\boldsymbol{\Psi}_d\}_1^D, \{\boldsymbol{L}_n\}_1^N, \{\hat{\boldsymbol{\nu}}_n\}_1^N, \{\hat{\sigma}_n \text{ or } \kappa_n\}_1^N\}$

3

4  **while** *not converged* **do**

5    $\nabla\mathcal{L} \leftarrow 0$

6    **for** *batch in batches* **do**

7

8       %For each of $M$ Monte Carlo samples

9       **for** $m = 1 : M$ **do**

10

11          % sample from approximate posterior $q(\boldsymbol{X})$

12          **for** $d = 1 : D$ **do**

13             $\boldsymbol{\eta}_d^{(m)} \sim \mathcal{N}(\boldsymbol{0}, \boldsymbol{I}_T)$

14             $\boldsymbol{k}_d^{\frac{1}{2}} = \sigma_{\frac{1}{2},d} \exp\left(-\frac{(\boldsymbol{t}-t_0)^2}{2\tau_{\frac{1}{2},d}^2}\right)$       `// single column of` $\boldsymbol{K}$

15             $\boldsymbol{x}_d^{(m)} = \text{Toeplitz\_mult}(\boldsymbol{k}_d^{\frac{1}{2}}, \boldsymbol{\nu}_d + \boldsymbol{C}\boldsymbol{\eta}_d^{(m)})$     `// Appendix F`

16          $\boldsymbol{X}_m = [\boldsymbol{x}_1^{(m)}; \ldots; \boldsymbol{x}_d^{(m)}]$

17

18          % compute $q(\boldsymbol{F})$ and $\mathbb{E}_{q(\boldsymbol{F})}[p(\boldsymbol{Y}|\boldsymbol{F})]$

19          $\hat{\boldsymbol{\mu}}_n = \boldsymbol{X}_m^\top \hat{\boldsymbol{\nu}}_n$       `// variational mean`

20          $\hat{\boldsymbol{\sigma}}_n^2 = \text{diag}\left(\boldsymbol{X}_m^T \boldsymbol{S} \boldsymbol{L}_n \boldsymbol{L}_n^\top \boldsymbol{S} \boldsymbol{X}_m\right)$

21          $\log p_{YF}^{(m)} = \sum_{n,t \in batch} \mathbb{E}_{\mathcal{N}(f_{nt}; \hat{\mu}_{nt}, \hat{\sigma}_{nt}^2)}[\log p(y_{nt}|f_{nt})]$   `// Appendix K`

22

23       % compute KL terms

24       $\text{KL}_x = \frac{size(batch)}{size(data)} \sum_d \text{KL}[q(\boldsymbol{x}_d)||p(\boldsymbol{x}_d)]$      `// Appendix F`

25       $\text{KL}_f = \frac{size(batch)}{size(data)} \sum_n \text{KL}[q(\boldsymbol{f}_n)||p(\boldsymbol{f}_n)]$      `// Appendix G`

26

27       % update gradient with batch gradient

28       $\tilde{\mathcal{L}} = \frac{1}{M} \sum_m \log p_{YF}^{(m)} - \text{KL}_x - \text{KL}_f$

29       $\nabla\mathcal{L} \leftarrow \nabla\mathcal{L} + \nabla\tilde{\mathcal{L}}$

30

31    % update parameters based on total gradients (we use Adam in practice)

32    $\theta \leftarrow \theta + \gamma\nabla\mathcal{L}$

---

Note that we need to sample the full trajectory $\boldsymbol{x}_d$ before subsampling for each batch due to the correlations introduced by $\boldsymbol{K}$. In practice, we run the optimization for 2000 passes over the full data which we found empirically lead to convergence of the ELBO. We used $M = 20$ Monte Carlo samples for each update step when fitting synthetic data and $M = 10$ for the primate data. For all models, $q(\boldsymbol{X})$ was initialized at the prior $p(\boldsymbol{X})$. The prior scale parameters were initialized as $s_d = \rho||\boldsymbol{c}_d||_2^2$ where $\boldsymbol{c}_d$ is the $d^{th}$ row of the factor matrix $\boldsymbol{C}$ found by factor analysis [12], and $\rho = 3$ was found empirically to give good convergence on the primate data. When using a Gaussian noise model, noise variances were initialized as the $\sigma_n^2$ found by factor analysis. For negative binomial noise models, we initialized $\kappa_n = \frac{1}{T} \sum_t y_{nt}$ which matches the mean of the distribution to the data for $f = 0$. Length scales $\tau$ were initialized at 200 ms for all latent dimensions for the primate data and at $\approx 80\%$ of the ground truth value for the synthetic data. Synthetic data was fitted on a single GPU with 8GB RAM. Primate data was fitted on a single GPU with 12GB RAM and took approximately 30 hours for a single model fit to the full dataset at 25 ms resolution. We also note that when fitting data with a Gaussian noise model, we mean-subtracted the original data, whereas we

include explicit mean parameters in the Poisson and negative binomial noise models since they are non-linear (c.f. Appendix K).

**Code availability**  A PyTorch implementation of bGPFA is provided on GitHub.

## M  Cross-validation and kinematic decoding

In this section, we describe the procedure for computing cross-validated errors in Figure 2, and performing kinematic decoding analyses in Figure 3. In these analyses, expectations over $\boldsymbol{X}$ were computed using the posterior mean of $q(\boldsymbol{X})$ and expectations over $\boldsymbol{F}$ were computed using Monte Carlo samples from $q(\boldsymbol{F})$.

**Prediction errors**  To compute cross-validated errors, we divide the time points into a training and a test set, $\mathcal{T}_{train} = \{t_1, t_2, ..., t_{T_{train}}\}$ and $\mathcal{T}_{test} = \{t_{T_{train}+1}, ..., T\}$, and similarly for the neurons $\mathcal{N}_{train}$ and $\mathcal{N}_{test}$. We also define $\mathcal{T}_{tot} = \mathcal{T}_{train} \bigcup \mathcal{T}_{test}$ and $\mathcal{N}_{tot} = \mathcal{N}_{train} \bigcup \mathcal{N}_{test}$. We first fit the generative parameters $\theta_{gen}$ of each model to data from all the neurons at the training time points using variational inference (taking $\theta$ to include the parameters $\phi$ of $q_\phi(\boldsymbol{F})$):

$$\theta_{gen} = \mathrm{argmax}_{\theta_{gen}} \left[ p(\boldsymbol{Y}_{\mathcal{N}_{tot}, \mathcal{T}_{train}} | \theta_{gen}) \right]. \tag{70}$$

We then fix the generative parameters and infer a distribution over latents from the training neurons recorded at all time points using a second pass of variational inference:

$$q(\boldsymbol{X}_{1:D, \mathcal{T}_{tot}} | \boldsymbol{Y}_{\mathcal{N}_{train}, \mathcal{T}_{tot}}, \theta_{gen}) \approx p(\boldsymbol{X}_{1:D, \mathcal{T}_{tot}} | \boldsymbol{Y}_{\mathcal{N}_{train}, \mathcal{T}_{tot}}, \theta_{gen}). \tag{71}$$

Finally we use the inferred latent states and generative parameters to predict the activity of the test neurons at the test time points

$$\hat{\boldsymbol{Y}}_{\mathcal{N}_{test}, \mathcal{T}_{test}} = \int \boldsymbol{Y} p(\boldsymbol{Y}_{\mathcal{N}_{test}, \mathcal{T}_{test}} | \boldsymbol{X}_{1:D, \mathcal{T}_{test}}, \theta_{gen}) q(\boldsymbol{X}_{1:D, \mathcal{T}_{test}} | \boldsymbol{Y}_{\mathcal{N}_{train}, \mathcal{T}_{tot}}, \theta_{gen}) d\boldsymbol{X}_{1:D, \mathcal{T}_{test}} \tag{72}$$

This allows us to compute a cross-validated predictive mean squared error as

$$\epsilon = \frac{1}{|\mathcal{N}_{test}| \, |\mathcal{T}_{test}|} ||\hat{\boldsymbol{Y}}_{\mathcal{N}_{test}, \mathcal{T}_{test}} - \boldsymbol{Y}_{\mathcal{N}_{test}, \mathcal{T}_{test}}||_2^2. \tag{73}$$

**Kinematic decoding**  For kinematic decoding analyses, we only considered the latents and behavior prior to a period of approximately 5 minutes where the monkey disengaged from the task (the first 1430 seconds; Section 3.3). Cursor positions in the x and y directions were first fitted with cubic splines and velocities extracted as the first derivative of these splines. To evaluate kinematic decoding performance, we followed Keshtkaran et al. [5] and computed the expected activity of all neurons at all time points under our model:

$$\hat{\boldsymbol{Y}} = \int \boldsymbol{Y} p(\boldsymbol{Y}|\boldsymbol{F}) q(\boldsymbol{F}|\boldsymbol{X}) q(\boldsymbol{X}|\boldsymbol{t}) d\boldsymbol{X} d\boldsymbol{F}. \tag{74}$$

For non-Gaussian noise models, this can be viewed as the first non-linear step of a decoding model from the latent states $\boldsymbol{X}$. We then performed 10-fold cross-validation where 90% of the data was used to fit a ridge regression model which was tested on the held-out 10% of the data. The regularization strength was determined using 10-fold cross-validation on the 90% training data. The predictive performance was computed as the mean across the 10 folds. Models were fitted and evaluated independently for the hand x and y velocities, and the final performance was computed as the mean variance accounted for across these two dimensions. Results in Section 3.2 are reported as mean $\pm$ standard error across 10 different splits of the data into folds used for cross-validation.