# OpenReview forum: "Scalable Bayesian GPFA with automatic relevance determination and discrete noise models"
_NeurIPS.cc/2021/Conference — NeurIPS 2021 Poster_

### Official Review · Reviewer_eNxg · 2021-07-13

**Rating:** 7
**Confidence:** 4

**Summary:**

The authors introduce a new variant of GPFA for neural data that includes an ARD prior over the mixing matrix to automatically learn dimensionality. The authors also develop specific inference methods for this work as well as exploit structure in the covariance matrix for stability/speed. The authors test their model on synthetic data to demonstrate that the bGPFA model correctly identifies true latent dimensionality and performs well in situations with Poisson and Negative Binomial noise.  The authors also show model performance on multi-region reaching data in monkey cortex and explore the presence of time delays in the model.

**Main Review:**

######

The authors have thoughtfully and thoroughly addressed all of my concerns. With the proposed edits, I think this paper is very appropriate for publication in this year's NeurIPS.

######

This paper is clearly written and the contributions, in particular the the ARD prior as well as specifying an additional variational distribution over the noise observation , is an important advancement in this class of models used in neuroscience settings. The author's work exploring the monkey data is an interesting scientific contribution as well. However, the authors are overstating the advancements on a few points in the manuscript, and this should be corrected.

For one thing, in the abstract the authors state that their approach "also allows for non-Gaussian noise models more appropriate for electrophysiological recordings", and in the intro the authors state "bGPFA also naturally extends to non-Gaussian data where it recovers ground truth parameters and latent trajectories." A reader here is left with the impression that a central contribution here is that GPFA has not been used in non-Gaussian settings, but the reality is that there are very many modern instances of GPFA with non-Gaussian noise. In fact, it is now more common to use Poisson GPFA than canonical GPFA for neuroscience. The authors cite two of these instances (Zhao and Park 2016 and Duncker and Sahani 2018), but there are recent inference advancements for GPFA for three discrete observation models -- Poisson, Negative-Binomial, and Binomial ( Keeley et al 2020 ICML* ), and GPFA with Poisson observations has been used in recent analysis for motor behavior (Zhao et al 2020^) and also recently used to dissect signal and noise representations in neural data (Keeley et al 2020 Neurips^*) .
 A more appropriate comparison for the evaluations in fig 2 would be one of these Poisson GPFA models, not just canonical GPFA, as we are dealing with neural data. (see Zhao and Park 2016 vLGP code here, which is quite user-friendly: https://github.com/catniplab/vlgp).

Moreover, the Fourier trick used with the Toeplitz structured covariance matrix is quite similar to what is done in Keeley et al 2020 Neurips^* which represented of the variational distribution and prior covariance matrix in the Fourier domain for scalability for Poisson-observation GPFA.

I find the rest of the manuscript, including the evaluation on real data  and the thorough appendix and available code, very practical and useful. The automatic selection of dimensionality is helpful, and the discussion comparing to GPLVMs quite interesting to the machine learning community. The new inference with whitened parametrization is a nice contribution, as well the proof of standard GPFA equivalence for their variational setting.

It would be nice to have some discussion as to how this method's inference compares to other approaches to GPFA with Poisson noise.

If this authors can better contextualize their achievements, and perhaps show some results with a comparison to a Poisson GPFA model (perhaps even their own, as they stated they have in their code?), I will gladly raise my score.

Papers references:

*Keeley, SL, et al. "Efficient non-conjugate Gaussian process factor models for spike count data using polynomial approximations." International Conference on Machine Learning. PMLR, 2020.

^Zhao, Yuan, et al. "Stimulus-choice (mis) alignment in primate area MT." PLoS computational biology 16.5 (2020): e1007614.

^*Keeley SL, Aoi MC, Yu Y, Smith SL, BR & Pillow JW (2020). Identifying signal and noise structure in neural population activity with Gaussian process factor models. Advances in Neural Information Processing Systems (NeurIPS) 33: 13795-13805.


**Time Spent Reviewing:**

5 hours

---

> ### Author Response · Authors · 2021-08-10
> **Response to reviewer eNxg**
>
> **General comments**
>
> > This paper is clearly written and the contributions, in particular the ARD prior as well as specifying an additional variational distribution over the noise observation , is an important advancement in this class of models used in neuroscience settings. The author's work exploring the monkey data is an interesting scientific contribution as well.
>
> We thank the reviewer for their encouraging comments, and we are glad that they found our method useful and our analyses interesting. We also appreciate the helpful comments and suggestions, and in particular the pointers to additional relevant literature.
>
> > The authors are overstating the advancements on a few points in the manuscript, and this should be corrected. (...) A reader here is left with the impression that a central contribution here is that GPFA has not been used in non-Gaussian settings, but the reality is that there are very many modern instances of GPFA with non-Gaussian noise
>
> We have now added an additional paragraph in the related works section to highlight and describe how non-conjugate likelihoods have previously been incorporated into latent variable models for neuroscience including the case of Poisson GPFA. We agree with the reviewer that our primary contribution lies instead in unifying these ideas with a Bayesian formulation that facilitates automatic relevance determination as well as with techniques from the GP literature that facilitate computational scalability. We have rephrased the abstract and introduction accordingly in our revised manuscript to better contextualize our work within the broader computational neuroscience literature.
>
> **Further discussions of previous work**
> > It would be nice to have some discussion as to how this method's inference compares to other approaches to GPFA with Poisson noise
>
> In addition to further numerical comparisons, we have now added to our related work section a more thorough discussion of the similarities between our approach and previous GPFA methods with non-conjugate likelihoods, including Keeley et al. (2020a, 2020b), Zhao and Park (2017), and Duncker & Sahani (2018).
>
> > The Fourier trick used with the Toeplitz structured covariance matrix is quite similar to what is done in Keeley et al 2020 Neurips^* which represented the variational distribution and prior covariance matrix in the Fourier domain for scalability for Poisson-observation GPFA.
>
> As the reviewer notes, our approach shares several conceptual ideas with the inference procedure developed by Keeley et al. (2020b) in that (i) both use variational inference to accommodate non-conjugate likelihoods, and (ii) both make use of Fourier transforms for computational efficiency. However, there are also some important differences.
> * Notably, while Keeley et al. perform inference in the Fourier domain where they assume a factorized variational distribution, we use the Fourier domain for an efficient parameterization of, and multiplication by, a circulant matrix which forms only one part of our variational posterior. However, in contrast to Keeley et al., we perform inference in the time domain and our variational posterior further includes (i) a diagonal matrix $\mathbf{\Psi}$, and (ii) the square root of the prior covariance $\mathbf{K}^{\frac12}$ (c.f. eqs 7 & 10; note that Fourier transforms are also used to compute $\mathbf{K}^{\frac12} \mathbf{v}$ products in $\mathcal{O}(T \log T)$ time).
> * This approach has the desirable properties of promoting smoothness in the posterior mean via $\mathbf{K}^{\frac12}$ and allowing for non-stationary posterior covariances via $\mathbf{\Psi}$ (c.f. Appendix E). To the best of our understanding, Keeley et al. instead use a variational posterior which factorizes across Fourier modes and therefore has a temporally constant marginal posterior variance.
>
>
> **Further method comparisons on synthetic and experimental data**
> > A more appropriate comparison for the evaluations in fig 2 would be one of these Poisson GPFA models, not just canonical GPFA, as we are dealing with neural data.
>
> In Figure 2a-b, one of our primary goals was to provide some intuition as to how Bayesian GPFA works and how it can be used for automatic relevance determination using the maximum likelihood principle, similar to Figure 4 of Bishop [5]. For this reason, we considered a simple Gaussian setting where vanilla GPFA corresponds to exact model recovery which provides a natural baseline. However, we agree that it is also interesting to compare performance on non-Gaussian data more similar to the biological recordings, and we do this in our revised manuscript.
> * We have now included an additional figure which uses synthetic spike count data to compare Poisson FA, P-GPFA and bGPFA with and without ARD (all of which are implemented as special cases of bGPFA in our codebase). Here we see similar results to Figure 2b where bGPFA again recapitulates the performance of P-GPFA without the need for cross-validation to select an appropriate latent dimensionality.
> * For a further comparison, we also include a similar analysis on the monkey reaching dataset. In this context of biological recordings, we observe similar results to the synthetic data where bGPFA with ARD again recovers the optimal latent dimensionality and performance of P-GPFA and bGPFA without ARD.
>
> >If the authors can better contextualize their achievements, and perhaps show some results with a comparison to a Poisson GPFA model (perhaps even their own, as they stated they have in their code?), I will gladly raise my score.
>
> We thank the reviewer for these suggestions and hope that our responses have provided some reassurance that our revised manuscript will address these questions in a satisfactory manner.

---

### Official Review · Reviewer_Lhbz · 2021-07-15

**Rating:** 7
**Confidence:** 4

**Summary:**

The authors propose a scalable inference algorithm for the GP factor analysis method and apply it for analysis of neural activity. The main advantages of the method are very fast execution, support for non-Gaussian likelihoods, and automatic selection of dimensionality for multivariate cases.

**Limitations And Societal Impact:**

Yes

**Main Review:**

The GPFA model is relevant for the field, especially for the application of neural activity that fits well within the traditional scope of NeurIPS. The main contribution is also very clear, providing tangible improvement for the computational cost for its inference and hence supporting inference for longer temporal time scales. The paper also includes additional contributions related to the noise model and automatic dimensionality estimation, but these are fairly standard elements that are easy to incorporate as part of models like this. Overall, the method has potential for becoming a standard tool for the application.

The technical solution, however, is quite straightforward and limits the originality of the work; I do not see significant contributions for the more broad inference literature. The ARD prior used for estimating the dimensionality is the standard choice most researchers would start from, given its prevalent use in other FA methods. The variational approximation is based on recent literature and uses well-justified non-trivial elements like the whitened parameterization of [18] and suitable parameterization for fast computation as in Eq. (10), but it still remains more as a case study of how to best to VI for this particular model. Nevertheless, the algorithm itself seems very reasonable with carefully chosen details for efficient computation.

The empirical experiments are good, including both a clear study on synthetic data and a highly detailed application in modeling neural activity. While the application case would be already a bit too detailed for most machine learning venues, it does fit NeurIPS well. The graphical illustrations are well designed.

Overall, the paper is well written, technically sound and non-trivial, and makes a clear contribution for a model of practical relevance. Even though the significance of the technical development outside the GPFA is a bit limited, I cannot think of any particular flaws in the presentation or execution.

**Time Spent Reviewing:**

0.75

---

> ### Author Response · Authors · 2021-08-10
> **Response to reviewer Lhbz**
>
> **General comments**
> > Overall, the paper is well written, technically sound and non-trivial, and makes a clear contribution for a model of practical relevance.
>
> We thank the reviewer for the kind comments and agree with the view that latent variable models such as GPFA and related methods are important for modern neuroscience. This is especially true as the field is recording from increasing numbers of neurons for increasing durations of time which challenges our existing computational paradigms while also further increasing the need for statistical tools to extract meaningful information from such large population recordings.
>
> We agree that none of the technical contributions of our Bayesian GPFA method are likely to revolutionize the inference literature but also remain of the opinion (as the reviewer seems to be) that it is important to develop methods that use existing ideas in practice to solve topical problems. Additionally, putting together several existing ideas can be non-trivial, and we highlight here our use of e.g. circulant-structured variational posteriors for inference over the latent trajectories. This parameterization has not previously been used in the VI literature but it is well-suited to the neuroscience setting of regularly binned spike counts, especially as the field moves towards more unconstrained experiments in freely moving animals with long continuous neural recordings.

---

### Official Review · Reviewer_6k1i · 2021-07-15

**Rating:** 5
**Confidence:** 4

**Summary:**


The authors propose a scalable inference method for Bayesian GPFA and show extensive results applying bGPFA to neural recordings from M1 and S1. They show interesting scientific findings of primate motor cortex with one reaching dataset.


**Limitations And Societal Impact:**

1. The data is from paper [25] which proposed a method called AutoLFADS. Their method is more advanced than GPFA and showed an extensive comparison with GPFA in [25]. I'm wondering whether bGPFA could compete against it to establish its modeling contribution. I think it might be hard but it's still worth showing. Cuz applying a well-known model to a well-known dataset and uncovering existing knowledge is not significant enough.

2. The main technical novelty in this paper is scalable inference. But the inference tricks exploited in this paper are fairly common and standard in the field of GP, e.g., considering eq 5 as a GPLVM with a linear kernel.

3. In the second experiment, the authors apply bGPFA to both M1 and S1 regions. The only conclusion is "this broad peak reflects the fact that these neural populations encode both current behavior in S1 as well as future behavior in M1". This seems to be a bit straightforward since the latent is found using both neural populations. What would be more interesting to show is what the latent would look like, how many latent dimensionalities are found using ARD (this is mentioned for individual fitting)? How the latent compares with the latent from M1 and S1 only? I think for latent dynamic/factor models for neural populations, it's more interesting to visualize more latents.





**Main Review:**


I think the last two points "motor preparation is important in the unconstrained self-paces task" and "slow dynamics reflect motivation or task engagement" are two novel findings in the neuroscience community (correct me if I'm wrong given my limited knowledge). But the motor preparation is only presented with a weak correlation, and the long timescale part has everything in the appendix. That kind of makes me doubt the true significance of these results. All other results are consistent with existing findings which are not strong scientific contributions. The modeling part is a bit weak given that neither bGPFA nor the scalable inference is novel, thus leading to an incremental contribution to the field of computational neuroscience.


**Time Spent Reviewing:**

2 hours

---

> ### Author Response · Authors · 2021-08-10
> **Response to reviewer 6k1i**
>
> **Response to main review**
> > I think the last two points "motor preparation is important in the unconstrained self-paced task" and "slow dynamics reflect motivation or task engagement" are two novel findings in the neuroscience community. (...) But the motor preparation is only presented with a weak correlation, and the long timescale part has everything in the appendix.
>
> We thank the reviewer for their comments and suggestions and agree that the findings of (i) explicit preparatory dynamics despite the unconstrained nature of the task, and (ii) long-timescale latent dimensions reflecting task engagement are interesting neuroscientific findings.
>
> * To address the concerns of the reviewer with respect to these findings, we first note that the observed correlation of $\rho \approx 0.4$ between the distance-to-prep in latent space and reaction time is comparable to previous findings by Afshar et al. (2011) in a trial-structured reaching task. We think $\rho \approx 0.4$ is quite a strong correlation considering that motor preparation in M1 is not the only factor affecting reaction times, which could also depend on e.g. task engagement (as we show in Appendix C), reach direction, propagation of sensory information in the visual hierarchy etc.
> * While we do not claim that the M1 latent state is the sole or necessarily primary contributor to variability in reaction times, we find it interesting that it is robustly predictive of reaction time since this validates the behavioral relevance of the latent variables identified by bGPFA and illustrates motor preparation despite such preparatory dynamics not being task-imposed.
>
> * We also agree that the long-timescale dynamics and task engagement are interesting scientific findings and now include the figure and discussion on this topic in the main text rather than leaving it in the appendix. We agree that this section strengthens the paper by further highlighting the types of scientific questions that can be answered using bGPFA.
>
> **Response to limitation 1**
> > The data is from paper [25] which proposed a method called AutoLFADS. Their method is more advanced than GPFA and showed an extensive comparison with GPFA in [25].
>
> The continuous reaching dataset was indeed recently analyzed in a preprint by Keshtkaran et al. [25] using Auto-LFADS after chunking the data into pseudo-trials for computational tractability (although the data was originally recorded by O’Doherty et al. [38]). We agree that LFADS is an intriguing method and a valuable alternative to many latent variable models employed in neuroscience. However, our primary interest is to learn interpretable latent trajectories and perform dimensionality reduction. In contrast, LFADS has a different objective, namely to learn a dynamical system and recover smoothened firing rates (although LFADS can of course be followed by e.g. PCA for dimensionality reduction). Additionally, (Auto-)LFADS has a much greater computational cost than bGPFA and requires access to a large computational cluster or extensive cloud computing resources. Finally, bGPFA is in many ways similar to P-GPFA and will likely fare similarly in a comparison with LFADS after optimizing for the latent dimensionality. For these reasons, we focused on non-parametric methods in our work and have now instead included further comparisons with vanilla GPFA, P-GPFA, and P-FA as suggested by several reviewers.
>
> **Response to limitation 2**
> > The main technical novelty in this paper is scalable inference. But the inference tricks exploited in this paper are fairly common and standard in the field of GP, e.g., considering eq 5 as a GPLVM with a linear kernel.
>
> We agree that formulating a Bayesian linear model as a linear Gaussian Process is not in itself novel, and neither is the use of ARD in linear models. However, combining this linear readout with a GP prior over latent trajectories leads to a deep GP with a structure which we exploit for scalability and non-conjugate noise models. In particular, we show that the linear GP in the observation model can be reformulated as a sparse variational GP ([17]) which facilitates non-conjugate likelihoods.
> We also separately develop a new variational framework for inferring the latent processes, involving a whitened parameterization with a carefully structured posterior covariance matrix and utilizing the Fourier domain for computational scalability. This framework exhibits good performance on long continuous recordings with $\mathcal{O}(T \log T)$ time complexity and outperforms e.g. low rank approximations to the posterior (Appendix E).
>
> **Response to limitation 3**
> > In the second experiment, the authors apply bGPFA to both M1 and S1 regions. The only conclusion is "this broad peak reflects the fact that these neural populations encode both current behavior in S1 as well as future behavior in M1". This seems to be a bit straightforward since the latent is found using both neural populations.
>
> We agree that the ‘broad peak’ in the decoding analysis from M1 and S1 recordings is not in itself of significant scientific interest.
> Instead, we find it interesting how decoding compares to the case where the M1 activity has been time-shifted relative to S1, which we have clarified in our revised manuscript.
> * In this case, the peak accuracy in kinematic reconstruction from latents becomes higher and narrower. This suggests that M1 and S1 contain synergistic information about kinematics, but that the biological properties of M1 and S1 must be taken into account for optimal decoding.
> * Additionally, our finding that the bGPFA ELBO is higher when taking this offset into account suggests the intriguing possibility that such representational delays between neural populations could be learned automatically in multi-region recordings to account for e.g. the propagation of information across the brain.
>
> > What would be more interesting to show is what the latent would look like, how many latent dimensionalities are found using ARD (this is mentioned for individual fitting)? How the latent compares with the latent from M1 and S1 only?
>
> As the reviewer notes, we fit bGPFA to M1 and S1 individually as well as to the M1+S1 data.
> * We find that bGPFA retains 20 dimensions for M1 and 13 dimensions for S1 while 22 dimensions (<< 20 + 13) are retained for the combined M1+S1 data (Appendix D). This is consistent with M1 and S1 containing both overlapping and non-redundant information which we now highlight in our submission.
> * When analyzing latent trajectories fitted to both M1 and S1 data, they tend to resemble the M1 latent trajectories most strongly since M1 neurons have the dominant contribution to most of the latent dimensions with highest information content.
> * For our analyses of latent dynamics, we originally focused on the M1 data for ease of comparison with prior literature. We have now repeated these analyses for the S1 data where we find poorer separation of latent trajectories across reach directions prior to movement onset ($z=0.42$ for S1 vs. $z=1.06$  for M1; c.f. Figure 4F). This result is consistent with the lower predictive decoding performance from S1, and we have included these S1 analyses in our revised manuscript.

---

### Official Review · Reviewer_sBjz · 2021-07-17

**Rating:** 6
**Confidence:** 3

**Summary:**

The authors introduce a Bayesian GPFA model, and extend this formulation to non-Gaussian noise models. A Gaussian prior over the loading matrix is proposed, and the paper employs a variational inference strategy for the inference of the model. The inference is shown to be efficient in the number of data points and helps in automatic relevance determination of the observations, as well as learning the appropriate number of latent variables to use. The method is used on a synthetic dataset to recover the dimensionality and the structure of the latent dimensions. Additionally, the method is applied to neural data from primates during a self-paced reaching task, with recordings from the primary somatosensory and motor cortices.

**Limitations And Societal Impact:**

Yes

**Main Review:**

The authors have a clear presentation of their ideas and the methodology. The implemented variational inference method performs well on synthetic data and on spiking neural data from primates. It also leads to interesting take-aways for the neural data during the different phases of the task.

Please include an explanation on why GPFA has a lower MSE than bGPFA on synthetic data in Figure 2b with latent dimensionality = 2 (and for most other values for latent dimensionality). If the algorithm is indeed more efficient in practice, please show this.

The analysis of primate recordings using comparable methods includes the comparison between the similarity matrices of preparatory dynamics computed using raw data and FA – please comment on these same similarity matrices computed using GPFA, perhaps fit on a smaller number of trials due to computational considerations. Please also comment on the dimensionality found in the neural data using bGPFA vs. other comparable methods (for example, GPFA using the relevant noise models). Finally, please comment on the structure in the latent dimensions (as in Figure 3e) using other comparable methods.

The authors stress the possibility to perform inference on discrete (spike count) data using the proposed method. However, they do not use this capability for the electrophysiological neural data, choosing to bin the data and analyze the firing rates instead. Please explain.

While the discoveries that are made on the neural data are interesting, further analysis may be needed to show the utility of the method against comparable methods.

**Time Spent Reviewing:**

2

---

> ### Author Response · Authors · 2021-08-10
> **Response to reviewer sBjz**
>
> > The authors have a clear presentation of their ideas and the methodology. The implemented variational inference method performs well on synthetic data and on spiking neural data from primates. It also leads to interesting take-aways for the neural data during the different phases of the task.
>
> We thank the reviewer for their comments and suggestions, and we are happy to hear that they found our neural data analyses interesting.
>
> **Performance on synthetic data**
>
> > ​​Please include an explanation on why GPFA has a lower MSE than bGPFA on synthetic data in Figure 2b with latent dimensionality = 2 (and for most other values for latent dimensionality). If the algorithm is indeed more efficient in practice, please show this.
>
> As noted in our general response, the synthetic data in Figure 2a-b is drawn from the GPFA generative model, and fitting GPFA thus corresponds to exact model recovery. We therefore do not in general expect other methods to outperform vanilla GPFA in this case. Instead, we use the synthetic example to illustrate how the maximum likelihood principle is used by bGPFA to automatically select the correct latent dimensionality, similar to Figure 4 of Chris Bishop’s seminal Bayesian PCA paper ([5]). We now also include a model comparison on synthetic spike count data where P-GPFA outperforms regular (GP)FA, and bGPFA again recovers the optimal P-GPFA performance without the need for cross-validation to infer an appropriate latent dimensionality.
>
> **Comparison with Poisson GPFA and Poisson FA on primate recordings**
> > Please also comment on the dimensionality found in the neural data using bGPFA vs. other comparable methods (for example, GPFA using the relevant noise models).
>
> * Similar to the synthetic data, we now include an analysis comparing the performance of Poisson FA, P-GPFA and bGPFA on the monkey reaching data for different latent dimensionalities. Here we show that bGPFA recovers both the optimal performance and latent dimensionality of the other methods without the need for explicit tuning of the latent dimensionality and with the capacity to scale to large datasets via our efficient variational inference framework.
> * In this more challenging setting of real experimental data, overfitting can also become more of an issue. As an example, bGPFA without ARD and P-GPFA perform similarly for their optimal latent dimensionality $D=4$ ($\mathcal{L}^{bGPFA}=0.4966$ vs. $\mathcal{L}^{PGPFA}=0.4964$). However, P-GPFA performs significantly worse when the latent dimensionality is overestimated ($\mathcal{L}^{bGPFA}=0.4973$ vs. $\mathcal{L}^{PGPFA}=0.4987$ for $D = 10$; an increase of 7e-4 vs 23e-4). In contrast to these two methods, bGPFA with ARD performs well while circumventing the need to select the latent dimensionality _a priori_ ($\mathcal{L}=0.4965$).
>
> **bGPFA captures more behaviorally relevant features than GPFA**
> > The analysis of primate recordings using comparable methods includes the comparison between the similarity matrices of preparatory dynamics computed using raw data and FA – please comment on these same similarity matrices computed using GPFA, perhaps fit on a smaller number of trials due to computational considerations.
>
> We have now used our variational inference framework to fit both Poisson GPFA and vanilla GPFA to the full monkey reaching dataset with a latent dimensionality matched to that inferred by bGPFA. We include an additional supplementary section in our revised manuscript where we compare the latent similarity matrices for these methods at target and movement onset to those found using bGPFA.
>
> Here we find slightly less structure in the pre-movement similarity matrix and thus less preparatory modulation by movement direction for P-GPFA ($z=0.87$; c.f. Figure 4f) and GPFA ($z=0.72$) than for bGPFA ($z=1.06$). However, these methods still exhibit more modulation than standard Factor Analysis ($z=0.50$). This suggests that bGPFA recovers latent trajectories which better capture key features of the behavioral output for this dataset.
>
> > Finally, please comment on the structure in the latent dimensions (as in Figure 3e) using other comparable methods.
>
> We now also compare the latent trajectories (similar to Figure 3e) for both FA and Poisson-GPFA. These show reduced modulation by movement direction and additionally exhibit a lack of temporal smoothness for FA due to the absence of an explicit smoothness prior as in the GPFA-based methods. Non-Bayesian GPFA also seems to generally infer faster timescales than bGPFA, possibly due to its increased susceptibility to short-timescale noise in the data.
>
> **Inference with discrete data**
> > The authors stress the possibility to perform inference on discrete (spike count) data using the proposed method. However, they do not use this capability for the electrophysiological neural data, choosing to bin the data and analyze the firing rates instead. Please explain.
>
> When we discuss ‘discrete’ data, we are referring to binned spike counts which take on non-negative integer values in contrast to modelling the data with a Gaussian distribution which can take any value in the real domain. This has been shown in a range of work to be important for recovering more meaningful latent trajectories (e.g. Zhang & Park 2017, Keeley et al. 2020b, Wu et al. 2017). We thank the reviewer for pointing out the potential for misunderstanding of this terminology and have clarified it in the revised paper. An interesting alternative to these discrete noise models would be to extend bGPFA to a point process noise model taking as input the actual real-valued spike times as in Duncker and Sahani (2018). This should be possible within our present inference paradigm, but we considered it beyond the scope of this particular submission.
>
>
> **General**
> > While the discoveries that are made on the neural data are interesting, further analysis may be needed to show the utility of the method against comparable methods
>
> We hope that our additional analyses and comparisons to FA, P-FA, GPFA and P-GPFA will help convince the reviewer of the utility of Bayesian GPFA for the computational neuroscience community.

---

### Author Response · Authors · 2021-08-10
**General response to reviewers**

We thank the reviewers for their time and suggestions which have been very helpful in improving our submission. Here we address some of the questions raised by several reviewers with further point-by-point responses in the individual comments.

**Technical contributions**\
Some reviewers comment on the novelty of bGPFA and the technical contributions of our submission.
* We agree with the reviewers that our primary contribution is not any single technical advance. Instead, our approach draws on many known techniques, including variational inference, whitened GP parameterizations, and ARD, while also building on a novel inference approach which we develop.
* However, we are of the opinion that an important task in machine learning and computational neuroscience is to leverage such ideas from basic machine learning research to solve practical problems - in this case to fit latent variable models with appropriate priors and noise models to large-scale continuous neural recordings with automatic inference of the latent dimensionality.
* We hope to have highlighted the importance of such models through our extensive analyses of experimental data - analyses that would have been difficult with most existing methods which do not readily scale to continuous recordings with tens of thousands of time bins. Additionally, we provide a ready-to-use PyTorch implementation of bGPFA which we hope will further facilitate its use by the broader computational neuroscience community.

**Variational inference approach**
* While most of our technical contributions build on existing work, we also introduce a new approach for variational inference with variational posteriors which leverage circulant matrices and Fourier transforms to ensure scalability and expressivity.
* We have now included an additional section in the discussion which considers how this differs from previous inference approaches in latent variable models with non-conjugate likelihoods. We also provide additional insights into these models by showing how our linear observation model with variational inference can be considered a special case of the stochastic variational GP (Hensman et al. 2015; Appendix G), and by formulating Bayesian GPFA within the context of GPLVMs and deep GPs which we hope can facilitate further research in the fields of linear and non-linear LVMs.

**Comparisons with other methods**\
Several reviewers comment on the comparison between bGPFA and GPFA on synthetic data in Figure 2a-b.
* In this figure, our primary goal was to illustrate how Bayesian GPFA uses the maximum likelihood principle to directly learn the latent dimensionality of the data, similar to Figure 4 of Bishop (1999). We used data generated from standard Gaussian GPFA and compared to this ‘true’ model in order to have a rigorous baseline and show that bGPFA approximately recovers the performance of standard GPFA with an optimized latent dimensionality. For this analysis, we did not expect bGPFA to exceed the performance of GPFA since GPFA is the true generative model and we used sufficient data that overfitting was not a significant issue.

In response to the reviewers’ comments, we now include additional analyses comparing performance on synthetic data drawn from a discrete model with Poisson noise. Here, bGPFA with a Poisson noise model exhibits similar performance to the optimal Poisson GPFA (P-GPFA) model without the need for cross-validation to select an appropriate latent dimensionality.
* For this dataset, the minimum negative predictive log likelihood ($\mathcal{L}$; lower is better) is $\mathcal{L} = 1.0629 \pm 0.0004$ for P-FA, $\mathcal{L} = 1.0479 \pm 0.0009$ for P-GPFA, and $\mathcal{L} = 1.0474 \pm 0.0002$ for bGPFA with ARD.
* We perform similar comparisons on a subset of the monkey reaching dataset and show that bGPFA again recovers the optimal dimensionality and performance of P-GPFA ($D^* \approx 4$ for this smaller dataset). For this analysis, we performed 10-fold cross-validation across 7 different dimensionalities with 5 random seeds for P-GPFA and bGPFA without ARD. This corresponds to a total of ~350 rounds of model training to infer the latent dimensionality for each method, which illustrates the practical utility of automatic relevance determination in this domain.
* We hope that our additional comparisons with P-GPFA and P-FA across datasets and dimensionalities will help convince the reviewers that our framework is useful for practical model fitting. We also note in passing that our variational inference framework and bGPFA implementation readily recover FA and GPFA with non-conjugate likelihoods as special cases, thus facilitating easy comparisons for different datasets which we hope will be useful for the community.

**References for responses to reviewers**\
Numbers in brackets correspond to reference numbers in our original submission.

Bishop (1999). “Bayesian PCA.” Advances in neural information processing systems. [5]

Duncker and Sahani (2018). “Temporal alignment and latent Gaussian process factor inference in population spike trains.” Advances in neural information processing systems. [11]

Hensman et al. (2015). “Scalable variational Gaussian process classification.” Proceedings of the 18th International Conference on Artificial Intelligence and Statistics. [17]

Keeley et al. (2020a). “Efficient non-conjugate Gaussian process factor models for spike count data using polynomial approximations.” Proceedings of the 37th International Conference on Machine Learning.

Keeley et al. (2020b). “Identifying signal and noise structure in neural population activity with Gaussian process factor models.” Advances in neural information processing systems.

Keshtkaran et al. (2021). “A large-scale neural network training framework for generalized estimation of single-trial population dynamics.” bioRxiv. [25]

O’Doherty et al. (2017). “Nonhuman primate reaching with multichannel sensorimotor cortex electrophysiology.” Zenodo. [38]

Wu et al. (2017). Gaussian process based nonlinear latent structure discovery in multivariate spike train data. Advances in neural information processing systems. [51]

Zhao et al. "Stimulus-choice (mis) alignment in primate area MT." PLoS computational biology.

Zhao & Park (2017). “Variational latent gaussian process for recovering single-trial dynamics from population spike trains.” Neural computation. [53]

---

> ### Author Response · Authors · 2021-08-31
> **Summary**
>
> We hope that the reviewers have had a chance to consider our additional experiments where we provide further comparisons to related methods and additional analyses of the primate reaching data. Additionally, we hope that our responses to the reviews have answered the questions that arose from our original submission in a satisfactory manner. Finally, we hope that these additions and clarifications have helped convince the reviewers of the utility of Bayesian GPFA for the neuroscience community, and we encourage the reviewers to let us know if they have any additional suggestions, concerns, or clarifying questions regarding either our original submission or our further experiments.

---

### Decision · Program_Chairs · 2021-09-27

**Decision:**

Accept (Poster)

**Comment:**

Two reviewers recommend acceptance and, after the discussion period, the other two reviewers don't oppose. I read the paper and I agree that although individual parts of the model and inference are known, it is their combination that brings significance to the work, especially by allowing its application to large scale data in neuroscience. I recommend Accept.